# ViP$^2$-CLIP: Visual-Perception Prompting with Unified Alignment for Zero-Shot Anomaly Detection

**Ziteng Yang**[*]                                                              *yangzt24@mails.tsinghua.edu.cn*
*Tsinghua University*

**Jingzehua Xu**[*]                                                              *xjzh23@berkeley.edu*
*Tsinghua University*

**Cong Liu**[*]                                                              *cong_l.lc@connect.hku.hk*
*The University of Hong Kong*

**Yanshu Li**                                                              *yanshu_li1@brown.edu*
*Brown University*

**Zepeng Li**                                                              *zp-li24@mails.tsinghua.edu.cn*
*Tsinghua University*

**Yeqiang Wang**                                                              *wangyeqianger@126.com*
*Shanghai Jiao Tong University*

**Xinghui Li**[†]                                                              *li.xinghui@sz.tsinghua.edu.cn*
*Tsinghua University*

**Reviewed on OpenReview:** *https://openreview.net/forum?id=KCRRuiQSIm*

## Abstract

Zero-Shot Anomaly Detection (ZSAD) aims to detect anomalies in a target dataset without any training samples, leveraging models trained on auxiliary data. While CLIP offers strong cross-modal representations for ZSAD, its pretraining objective inherently emphasizes global foreground semantics over fine-grained local defects. Consequently, its anomaly localization remains highly sensitive to prompt wording, limiting the effectiveness of existing methods that rely on explicit category labels. To overcome this limitation, we introduce ViP$^2$-CLIP, a lightweight CLIP-based ZSAD framework featuring Visual-Perception Prompting (ViP-Prompt) and Unified Text-Patch Alignment (UTPA). ViP-Prompt replaces fixed class-name tokens with image-conditioned cues to adaptively generate fine-grained prompts, obviating the need for manual templates and class-name priors. Furthermore, UTPA enforces a unified text-patch alignment strategy across multiple visual scales, jointly optimizing image-level detection and pixel-level localization. These mechanisms enable the model to precisely localize abnormal regions, exhibiting particular robustness in scenarios with ambiguous or privacy-constrained category labels. Extensive experiments on 14 industrial and medical benchmarks demonstrate that ViP$^2$-CLIP achieves competitive performance against existing state-of-the-art approaches, with particular strengths in pixel-level localization capability. Code is available at: https://github.com/kim0806/ViP2-CLIP.

---

   * Equal contribution. † Corresponding author.

# 1 Introduction

Large-scale vision-language models (VLMs) such as CLIP (Radford et al., 2021) exhibit remarkable zero-shot recognition capabilities by pretraining on massive image-text pairs. Leveraging these powerful cross-modal representations, recent works have successfully adapted CLIP to Zero-Shot Anomaly Detection (ZSAD) (Jeong et al., 2023), offering a practical solution for localizing irregularities without target-domain training data. However, unlike conventional classification that focuses on global foreground semantics, ZSAD inherently targets fine-grained, localized abnormalities. As illustrated in Fig. 1(a), existing CLIP-based methods typically formulate this by constructing 'normal' and 'anomalous' text prompts. Both image and text features are projected into a shared embedding space, where global visual features are aligned with prompts to estimate image-level anomaly scores, and local patch features are matched to produce pixel-level anomaly maps.

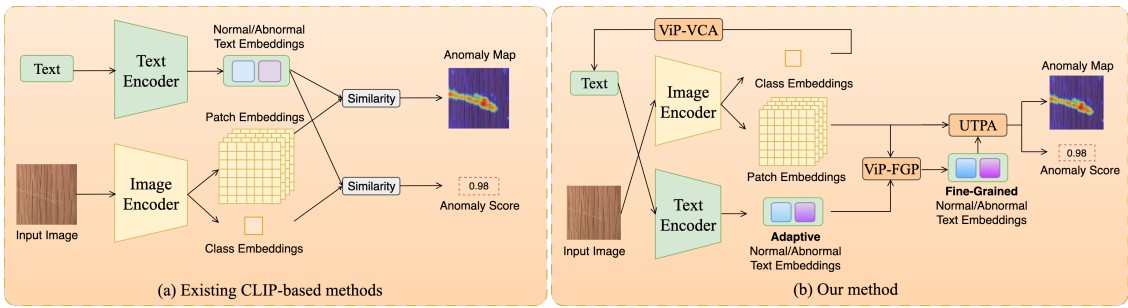

Figure 1: **Comparison between prior CLIP-based methods and ViP²-CLIP.** ViP²-CLIP replaces fixed class-name tokens with image-conditioned prompts and further incorporates a unified alignment strategy within training-based CLIP methods.

Recent state-of-the-art approaches primarily enhance ZSAD via prompt engineering, utilizing either hand-crafted templates or learnable tokens. Methods such as WinCLIP (Jeong et al., 2023), APRIL-GAN (Chen et al., 2023), AdaCLIP (Cao et al., 2024), and AA-CLIP (Ma et al., 2025a) incorporate explicit class names into prompts to boost discriminative power. However, we empirically observe that CLIP's segmentation quality remains highly sensitive to minor variations in class-name wording, even after auxiliary fine-tuning (see Appendix A for details). This sensitivity severely limits their applicability in real-world scenarios where category labels are ambiguous or privacy-constrained. While AnomalyCLIP (Zhou et al., 2024) mitigates this issue using object-agnostic learnable prompts, its uniform descriptive tokens struggle to adapt to complex and diverse visual patterns. Other methods like VCP-CLIP (Qu et al., 2024) and FAPrompt (Zhu et al., 2025) explore dynamic, visually guided prompt learning to bypass class labels. Nevertheless, these approaches often introduce additional architectural complexity and may suffer from an imbalanced trade-off between image-level detection and pixel-level localization performance.

To address these limitations, we propose ViP²-CLIP, a CLIP-based framework built upon two core components: Visual-Perception Prompting (ViP-Prompt) and a Unified Text-Patch Alignment (UTPA) strategy, as illustrated in Fig. 1(b). ViP-Prompt consists of a Visual-Conditioned Adapter (ViP-VCA) and a Fine-Grained Perception module (ViP-FGP). The ViP-VCA injects global visual cues into the prompt embedding space to facilitate semantic alignment, while the ViP-FGP aggregates multi-scale patch features to capture fine-grained irregularities. By replacing fixed class-name tokens with image-conditioned prompts that fuse both global and local visual features, ViP-Prompt reduces the reliance on manually specified category names and improves the stability of cross-modal alignment under ambiguous or inconsistent category descriptions. Building upon this, UTPA further enhances the framework by aligning these image-conditioned prompts with multi-scale patch features through a consistent alignment objective, enabling accurate anomaly detection and localization. Additional analyses in Section A.2 further demonstrate the robustness of the proposed framework under semantically equivalent category-name substitutions. Our key contributions are summarized as follows:

- We propose ViP²-CLIP, a lightweight CLIP-based ZSAD framework tailored for scenarios with ambiguous or privacy-constrained category labels. It achieves competitive performance with favorable computational efficiency across 14 diverse industrial and medical benchmarks.

- We design ViP-Prompt, which adaptively fuses global and local visual cues to generate image-conditioned prompts. This mechanism obviates the need for handcrafted templates and class-name priors, yielding more robust prompt representations.

- We introduce UTPA, a simple yet effective alignment strategy that incorporates a consistent text–patch alignment mechanism into training-based CLIP models, effectively mitigating the optimization conflict between image-level detection and pixel-level localization.

## 2 Related Work

**Zero-Shot Anomaly Detection (ZSAD)**  Zero-Shot Anomaly Detection aims to localize anomalous regions without access to target-domain training data, relying solely on auxiliary data. CLIP (Radford et al., 2021) advances this task by framing it as an image-text matching problem. Early methods, such as WinCLIP (Jeong et al., 2023), rely on handcrafted prompt ensembles, which require domain knowledge and provides limited semantic coverage. To overcome this, AnomalyCLIP (Zhou et al., 2024) introduces learnable, object-agnostic prompts. While this improves cross-domain generalization, static prompts still struggle to capture the variability of complex visual instances. This limitation has motivated the development of dynamic, visually guided prompt modeling.

Recent works primarily explore two directions for visual guidance. One line of work leverages visual features to guide prompt adaptation. For instance, AdaCLIP (Cao et al., 2024) employs hybrid prompts to enhance text-image alignment. GenCLIP (Kim et al., 2026) leverages general prompts through multi-layer prompting and dual-branch inference, improving both generalization and category-specific representation. AA-CLIP (Ma et al., 2025a) improves text embedding discriminability through a two-stage optimization, while ACD-CLIP (Ma et al., 2025b) generates level-specific textual descriptors from visual features. Despite their effectiveness, these approaches still rely on explicit class-name labels. As a result, their segmentation performance remains highly sensitive to wording variations, a problem that limited auxiliary fine-tuning cannot fully resolve. The second line of research eliminates category names entirely, using visual representations for adaptive prompt modeling. VCP-CLIP (Qu et al., 2024) projects global visual features into a handcrafted prompt space to improve segmentation. Similarly, FAPrompt (Zhu et al., 2025) learns fine-grained prompts conditioned on local visual priors from test images. While these approaches improve adaptability to ambiguous categories, they introduce substantial architectural complexity. Moreover, such methods often exhibit imbalanced performance, struggling to simultaneously optimize image-level detection and pixel-level localization.

In contrast, our method adopts a category-agnostic, visually guided prompt mechanism using only two lightweight adapters, avoiding the architectural overhead of prior methods. Furthermore, by introducing a unified text-patch alignment strategy, it mitigates the optimization conflicts inherent in traditional dual-branch designs, achieving robust and balanced performance for both detection and localization.

**Prompt Learning**  Prompt learning was initially proposed in natural language processing to enhance the adaptability of pre-trained models (Ouyang et al., 2022; Touvron et al., 2023) to diverse downstream tasks. CoOp (Radford et al., 2021) first introduced this paradigm to the vision domain by inserting trainable tokens into text inputs, allowing CLIP to adapt to specific tasks without requiring full model fine-tuning. However, static prompts often exhibit limited generalization on unseen classes. To enhance transferability, methods like CoCoOp (Zhou et al., 2022) and DenseCLIP (Rao et al., 2022) generate image-conditioned prompts that dynamically adapt to visual contexts. To further prevent overfitting and improve unseen class recognition, approaches such as KgCoOp (Yao et al., 2023) and TCP (Yao et al., 2024) regularize learnable tokens toward handcrafted templates. Alternatively, methods including MaPLe (Khattak et al., 2023a), PromptSRC (Khattak et al., 2023b), and MMRL (Guo and Gu, 2025) extend prompting to both modalities, jointly optimizing the image and text spaces to achieve deeper cross-modal alignment. More recent studies explore richer prompt parameterizations and learning paradigms. For instance, SurPL (Liu et al., 2025) introduces a surrogate latent space to improve optimization efficiency and prompt diversity, while VaMP (Cheng and Han, 2025) models prompts as distributions to better handle multi-modal uncertainty. CaPL (Gao and Dong, 2025) incorporates causal intervention to mitigate spurious correlations in vision-language alignment, thereby enhancing robustness under distribution shifts.

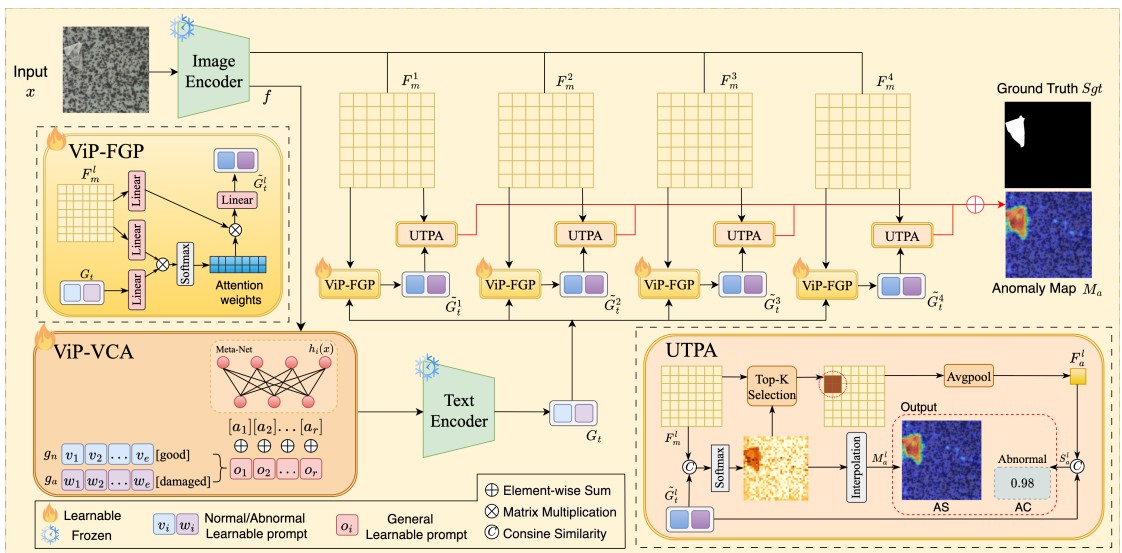

Figure 2: **Overview of ViP²-CLIP.** ViP-Prompt enhances cross-modal alignment by injecting global visual context (ViP-VCA) and integrating multi-scale patch features (ViP-FGP). Built on top of it, UTPA performs a unified alignment strategy across multiple layers, supporting both image-level detection and pixel-level localization.

## 3 Preliminary

CLIP consists of a text encoder $T(\cdot)$ and a visual encoder $F(\cdot)$, both implemented as multi-layer networks. By leveraging contrastive learning on large-scale image-text pairs, CLIP achieves superior zero-shot recognition capabilities. Given a class name $c$, we combine it with a text prompt template $G$ (e.g., 'A photo of a [cls]', where [cls] represents $c$), the resulting text is then fed into the text encoder $T(\cdot)$ to obtain the prompt embedding $g_c = T\big(G(c)\big) \in \mathbb{R}^D$. For an input image $x_i$, the visual encoder generates its global visual embedding $f_i \in \mathbb{R}^D$ and the local patch embeddings $f_i^m \in \mathbb{R}^{H \times W \times D}$. Specifically, given a category set $C$, CLIP computes the probability of image $x_i$ belonging to $c$ as follows:

$$p\big(y = c \mid x_i\big) = P(g_c, f_i) = \frac{\exp\big(\langle g_c, f_i \rangle / \tau\big)}{\sum_{c' \in C} \exp\big(\langle g_{c'}, f_i \rangle / \tau\big)}, \tag{1}$$

where $\tau$ is a temperature hyperparameter and $\langle \cdot, \cdot \rangle$ denotes the cosine similarity.

Unlike conventional classification, ZSAD flags deviations from normality rather than assigning foreground semantics. Therefore, most existing approaches (Jeong et al., 2023) instantiate two distinct text prompts: a normal prompt $g_n$ and an abnormal prompt $g_a$. At the image level, the anomaly score is defined as the similarity between the global feature $f_i$ and the abnormal prompt, computed as $P(g_a, f_i)$. At the pixel level, for each spatial location $(j, k)$, the model extracts the corresponding patch token $f_i^m(j, k)$. It then computes the local normal score $S_n(j, k) = P\big(g_n, f_i^m(j, k)\big)$ and anomaly score $S_a(j, k) = P\big(g_a, f_i^m(j, k)\big)$, which are subsequently used to construct a pixel-wise anomaly map.

## 4 Method

Fig. 2 illustrates the overall architecture of ViP²-CLIP. It first employs Visual-Perception Prompting (ViP-Prompt) (Section 4.1) to fuse learnable prompts with global and local visual features. This enables the prompts to adaptively capture fine-grained patterns and strengthen cross-modal alignment. Furthermore, a Unified Text-Patch Alignment (UTPA) strategy (Section 4.2) is introduced to jointly optimize the alignment between prompt tokens and multi-scale patch features. By aggregating these alignment signals across multiple layers, ViP²-CLIP achieves accurate image-level detection and pixel-level localization simultaneously.

### 4.1 Visual-Perception Prompting (ViP-Prompt)

ViP-Prompt leverages a Visual-Conditioned Adapter (ViP-VCA) and a Fine-Grained Perception module (ViP-FGP) to generate multi-level descriptive prompts. These prompts flexibly track the visual patterns of target objects, thereby enhancing CLIP's capability for ZSAD.

**Visual-Conditioned Adapter (ViP-VCA)**   To eliminate the dependence on handcrafted templates and class-name priors, we first define static learnable prompt templates for both normal and anomaly classes:

$$g_n = [v_1][v_2]\dots[v_e] \text{ good } [o_1]\dots[o_r], \tag{2}$$

$$g_a = [w_1][w_2]\dots[w_e] \text{ damaged } [o_1]\dots[o_r], \tag{3}$$

here, $\{v_i\}, \{w_i\} \in \mathbb{R}^C$ denote the learnable normal and anomalous vectors, respectively. The term $\{o_i\} \in \mathbb{R}^C$ represents generic learnable tokens that replace explicit class labels. We adopt the adjectives 'good' and 'damaged' to guide the prompts in learning richer normal and anomalous semantics. By autonomously optimizing $g_n$ and $g_a$, the model can capture generic anomaly patterns across diverse objects.

To further condition these prompts on the target object and adapt to complex detection scenarios, we introduce a lightweight Meta-Net $h_i(\cdot)$ (Linear-ReLU-Linear). This network maps the global visual embedding $f$ extracted by the image encoder into the text embedding space, yielding dynamic tokens $\{a_i\}_{i=1}^r = h_i(f)$, where $a_i \in \mathbb{R}^C$ and $r$ is the number of mapped tokens. These dynamic tokens are then fused with the static generic tokens to generate object-aware prompts. Thus, the final prompt structure is defined as follows:

$$g_n = [v_1][v_2]\dots[v_e] \text{ good } [z_1]\dots[z_r], \tag{4}$$

$$g_a = [w_1][w_2]\dots[w_e] \text{ damaged } [z_1]\dots[z_r], \tag{5}$$

where $z_i = o_i + a_i$. This prompt structure is robust and object-adaptive, regardless of the specific descriptive tokens used, as further demonstrated in Section 5.

**Fine-Grained Perception Module (ViP-FGP)**   To enhance the fine-grained perceptual capacity of the prompts, we introduce an attention-based interaction module between the text prompts and multi-scale visual features. Specifically, we project both the text embeddings and the local patch embeddings into a shared $C$-dimensional space. Let $G_t \in \mathbb{R}^{2\times C}$ denote the prompt embeddings from the text encoder, and $F_m^l \in \mathbb{R}^{HW \times D}$ represent the smoothed local visual embeddings from the $l$-th layer of the visual encoder. We apply three learnable linear mappings to produce the query $Q_t = G_t W_q$, key $K_m^l = F_m^l W_k$, and value $V_m^l = F_m^l W_v$, where $W_q \in \mathbb{R}^{C \times C}$ and $W_k, W_v \in \mathbb{R}^{D \times C}$ are projection matrices.

Next, we compute the attention weights via $\text{Softmax}(Q_t K_m^{l\top}/\sqrt{C})$ and apply them to $V_m^l$, yielding the enhanced token embeddings. Finally, a linear projection $W_o \in \mathbb{R}^{C \times D}$ transforms these embeddings into layer-specific prompts $\tilde{G}_t^l$, which seamlessly integrate fine-grained local visual cues:

$$\tilde{G}_t^l = \left(\text{Softmax}(Q_t K_m^{l\top}/\sqrt{C})\, V_m^l\right) W_o, \tag{6}$$

where $\tilde{G}_t^l \in \mathbb{R}^{2 \times D}$ comprises the fine-grained normal prompt $\tilde{g}_n$ and the anomalous prompt $\tilde{g}_a$.

To evaluate the fine-grained sensitivity of the ViP-FGP module, Figure 3 visualizes the attention maps of both normal and anomalous prompts across diverse images. The normal prompts naturally attend to regular textures and structures, whereas the anomalous prompts precisely concentrate on defects and irregular regions, demonstrating a high sensitivity to anomalous patterns. These results confirm that the ViP-FGP module effectively refines prompt embeddings by incorporating localized visual priors, thereby significantly strengthening cross-modal alignment.

### 4.2 Unified Text-Patch Alignment (UTPA)

Recent CLIP-based ZSAD models typically adopt a dual-branch alignment scheme (Chen et al., 2024; Zhou et al., 2024): one branch aligns prompts with global visual embeddings for image-level detection, while

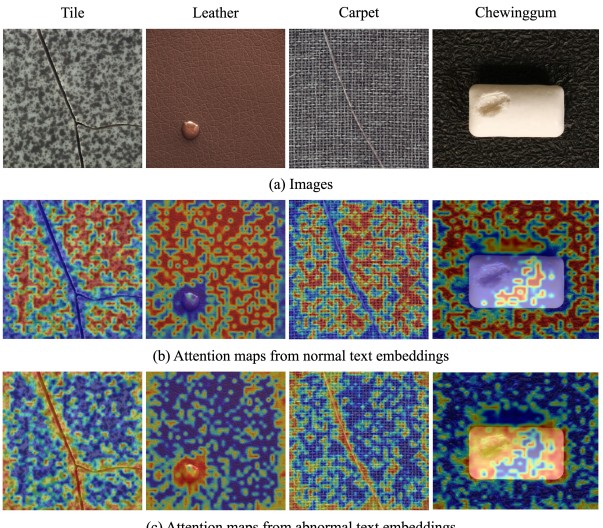

Tile  Leather  Carpet  Chewinggum

(a) Images

(b) Attention maps from normal text embeddings

(c) Attention maps from abnormal text embeddings

Figure 3: **Visualization of attention maps from different prompts in the ViP-FGP module.** Regions with higher attention are highlighted in red. Normal prompts focus on regular patterns, whereas anomalous prompts accurately localize defect regions, demonstrating the effectiveness of ViP-FGP in refining prompt representations.

the other aligns them with local patch embeddings for pixel-level localization. However, operating in two separate feature subspaces often introduces optimization conflicts, making it difficult to achieve high-precision detection and localization simultaneously.

To address this limitation, we introduce the UTPA strategy, which formulates anomaly detection and localization as a unified alignment task between text embeddings and local patch features. Concretely, to obtain the pixel-level anomaly map, we follow the standard practice: for the $l$-th layer feature map $F_m^l$, the patch token at spatial location $(j, k)$, denoted as $f_{j,k}^l \in \mathbb{R}^D$, receives its anomaly score by computing the similarity $P(\tilde{g}_a, f_{j,k}^l)$, as defined in equation 1. Subsequently, we upsample the anomaly scores of all patch tokens to produce the anomaly map $M_a^l$:

$$M_a^l = \mathrm{Up}\big(P(\tilde{g}_a, F_m^l)\big), \tag{7}$$

where $Up(\cdot)$ denotes channel-wise upsampling operation.

Beyond pixel-level localization, patch-wise anomaly responses can also be aggregated to derive image-level anomaly scores. Such aggregation principles have been widely studied in multiple instance learning (MIL) (Ilse et al., 2018), weakly supervised object localization (WSOL) (Zhou et al., 2015), and patch-based anomaly detection methods such as PatchCore (Roth et al., 2021), where image-level predictions are inferred from regional responses rather than global representations. Inspired by these studies, we also derive image-level anomaly scores from local patch representations rather than introducing an additional global alignment branch (Chen et al., 2024; Zhou et al., 2024). Since image-level abnormality is often dominated by highly anomalous regions, we select the Top-$K$ patch tokens with the highest $P(\tilde{g}_a, f_{j,k}^l)$ values at the $l$-th layer. The representative anomaly feature $F_a^l$ is then obtained by average pooling over the selected patches:

$$F_a^l = \frac{1}{K} \sum_{(j,k) \in \mathcal{T}_l} f_{j,k}^l, \tag{8}$$

where $\mathcal{T}_l$ denotes the index set of the selected Top-$K$ patches. This Top-$K$ pooling mechanism explicitly directs the model's attention to the most suspicious regions, ensuring that the image-level representation faithfully reflects localized defects. The image-level anomaly score $S_a^l$ at the $l$-th layer is then computed as $S_a^l = P(\tilde{g}_a, F_a^l)$. By enforcing this unified text-patch alignment strategy, UTPA effectively reduces the inconsistency between detection and localization objectives, leading to more balanced ZSAD performance.

### 4.3 Training and Inference

During training, ViP$^2$-CLIP is optimized by minimizing a combined objective comprising a global loss $L_{\text{global}}$ and a local loss $L_{\text{local}}$ (Zhou et al., 2024):

$$L_{\text{total}} = \sum_{l=1}^{N} L_{\text{global}}^{S^l} + \lambda \sum_{l=1}^{N} L_{\text{local}}^{M^l}, \tag{9}$$

where $\lambda$ is a balancing hyperparameter and $N$ represents the number of intermediate layers used for alignment. The global loss $L_{\text{global}}$ is formulated as a cross-entropy loss that maximizes the cosine similarity between the corresponding text embeddings ($\tilde{g}_n$ or $\tilde{g}_a$) and the representative global visual feature $F_a^l$. Meanwhile, the local loss $L_{\text{local}}$ jointly optimizes pixel-level alignment by combining a Focal loss (Lin et al., 2017) and a Dice loss (Li et al., 2019):

$$\begin{aligned} L_{\text{local}} = \text{Focal}\big(\text{Up}([M_n^l, M_a^l]), S_{\text{gt}}\big) \\ + \text{Dice}\big(\text{Up}(M_n^l),\, I - S_{\text{gt}}\big) \\ + \text{Dice}\big(\text{Up}(M_a^l),\, S_{\text{gt}}\big), \end{aligned} \tag{10}$$

where $[\cdot, \cdot]$ denotes channel-wise concatenation. For each layer $l$, $M_n^l$ and $M_a^l$ are the predicted normal and anomaly score maps, $S_{\text{gt}}$ is the ground-truth mask, and $I$ is an all-ones matrix of the same dimensions.

During inference, the final image-level anomaly score is calculated by averaging the scores across all selected layers: $\text{Score} = \frac{1}{N} \sum_{l=1}^{N} S_a^l$. For pixel-level prediction, we aggregate the intermediate maps $M_n^l$ and $M_a^l$ to compute the final spatial anomaly map $\text{Map} \in \mathbb{R}^{H \times W}$ as follows:

$$\text{Map} = G_\sigma\left( \frac{1}{N} \sum_{l=1}^{N} \Big( \tfrac{1}{2}\big(I - \text{Up}(M_n^l)\big) + \tfrac{1}{2}\text{Up}(M_a^l) \Big) \right), \tag{11}$$

where $G_\sigma$ denotes a Gaussian smoothing filter applied to refine the prediction boundaries.

## 5 Experiments

### 5.1 Setup

**Datasets & Baselines** To comprehensively evaluate ViP$^2$-CLIP across diverse scenarios, we conduct extensive experiments on 14 public benchmarks spanning both industrial and medical domains. For industrial defect detection, we utilize MVTec AD (Bergmann et al., 2019), VisA (Zou et al., 2022), MPDD (Jezek et al., 2021), BTAD (Mishra et al., 2021), KSDD (Tabernik et al., 2020), DAGM (Wieler and Hahn, 2007), and DTD-Synthetic (Aota et al., 2023). For medical anomaly detection, we evaluate on HeadCT (Salehi et al., 2021), BrainMRI (Salehi et al., 2021), Brain35H (Hamada, 2020), ISIC (Codella et al., 2018), CVC-ClinicDB (Bernal et al., 2015), CVC-ColonDB (Tajbakhsh et al., 2015), and Kvasir (Jha et al., 2019). We benchmark our method against six representative and reproducible ZSAD approaches: WinCLIP (Jeong et al., 2023), APRIL-GAN (Chen et al., 2023), AnomalyCLIP (Zhou et al., 2024), AdaCLIP (Cao et al., 2024), AA-CLIP (Ma et al., 2025a), and FAPrompt (Zhu et al., 2025).

**Metrics** We adopt standard evaluation metrics for a fair comparison. For image-level anomaly detection, we report the Area Under the Receiver Operating Characteristic curve (AUROC), Average Precision (AP), and maximum F1-score (F1). For pixel-level anomaly segmentation, we report AUROC, the Area Under the Per-Region Overlap curve (AUPRO), and F1.

**Implementation Details** We adopt the publicly released CLIP (ViT-L/14@336px) as our frozen feature extractor. The learnable prompts consist of 10 tokens in total, 3 of which are dynamically conditioned on the global visual embedding. For the UTPA strategy, we select the top 50 anomalous patches per layer as image-level descriptors, and perform cross-modal alignment at intermediate layers 6, 12, 18, and 24 (Zhou et al., 2024; Cao et al., 2024). Following standard zero-shot protocols, we adopt a cross-dataset

Table 1: **ZSAD performance comparison on industrial domain.** This table reports the performance (%) of different methods on seven industrial datasets, evaluated at both image-level detection (AUROC, AP, F1) and pixel-level location (AUROC, PRO, F1). The best performance is shown in bold, with the second-best underlined. Multi-seed results and standard deviations are provided in Appendix C.

| Task | Datasets | $|\mathcal{C}|$ | WinCLIP | APRIL-GAN | AnomalyCLIP | AdaCLIP | AA-CLIP | FAPrompt | ViP²-CLIP |
|---|---|---|---|---|---|---|---|---|---|
| | | | *CVPR 2023* | *CVPRw 2023* | *ICLR 2024* | *ECCV 2024* | *CVPR 2025* | *ICCV 2025* | *-* |
| | MVTec AD | 15 | (**91.8**, **96.5**, **92.7**) | (86.1, 93.5, 90.4) | (91.6, 96.4, **92.7**) | (90.1, 95.6, 92.3) | (89.1, 94.7, 90.0) | (**91.8**, 95.6, 92.3) | (91.2, 96.0, 92.0) |
| | VisA | 12 | (78.1, 81.2, 78.2) | (77.5, 80.9, 78.7) | (82.0, 85.3, 80.4) | (87.2, 89.7, 83.5) | (78.9, 82.3, 78.8) | (85.0, 86.7, 82.6) | (**88.5**, **90.4**, **84.8**) |
| Image-level | MPDD | 6 | (61.5, 69.2, 77.5) | (76.8, 83.0, 81.0) | (77.5, 82.5, 80.4) | (74.8, 78.6, **83.3**) | (56.9, 66.5, 74.5) | (**79.7**, 83.0, 82.3) | (79.7, **84.5**, 82.4) |
| (AUROC, AP, F1) | KSDD | 1 | (92.4, 82.9, 77.7) | (96.5, 91.2, 85.4) | (97.8, 94.2, 89.7) | (97.2, 92.6, 89.5) | (96.0, 89.2, 84.7) | (**98.6**, **96.4**, **93.6**) | (98.1, 95.8, 93.2) |
| | BTAD | 3 | (68.2, 70.9, 67.8) | (73.7, 69.7, 68.2) | (88.2, 88.2, 83.8) | (89.3, 96.5, 90.9) | (93.4, 97.6, 93.9) | (92.5, 93.7, 89.9) | (**95.0**, **98.4**, **94.7**) |
| | DAGM | 10 | (91.8, 79.5, 75.7) | (94.4, 83.9, 80.2) | (97.7, 92.4, 90.1) | (98.2, 92.3, 90.9) | (95.3, 87.5, 84.5) | (98.4, 95.1, 92.3) | (**98.5**, 94.3, **92.6**) |
| | DTD-Synthetic | 12 | (95.1, 97.7, 94.1) | (85.6, 94.0, 89.1) | (93.9, 97.2, 93.6) | (**96.3**, **98.1**, **95.5**) | (94.0, 98.0, 94.3) | (96.1, 98.1, 95.1) | (95.5, 98.1, 94.3) |
| | AVERAGE | - | (82.1, 82.1, 80.5) | (84.4, 85.2, 81.9) | (89.8, 90.9, 87.2) | (90.4, 91.9, 89.4) | (86.7, 88.0, 86.5) | (91.7, 92.7, 89.7) | (**92.4**, **93.9**, **90.6**) |
| | MVTec AD | 15 | (85.1, 64.6, 24.8) | (87.6, 44.0, 43.3) | (91.1, 81.4, 39.1) | (89.6, 37.8, 45.1) | (**91.5**, 86.5, **46.7**) | (89.4, 85.1, 40.8) | (90.5, **87.1**, 43.1) |
| | VisA | 12 | (79.6, 56.8, 9.0) | (94.2, 86.6, 32.3) | (95.5, 86.7, 28.3) | (95.5, 56.8, **37.0**) | (94.7, 82.7, 29.4) | (95.8, 88.6, 28.8) | (95.4, **92.2**, 33.6) |
| | MPDD | 6 | (71.2, 40.5, 15.4) | (94.3, 83.8, 31.3) | (96.5, 88.7, 34.2) | (96.1, 60.3, 31.9) | (96.0, 86.6, 26.5) | (96.3, 88.3, 34.8) | (**97.2**, **92.6**, **35.9**) |
| Pixel-level | KSDD | 1 | (92.8, 70.3, 15.8) | (93.2, 84.1, 43.6) | (98.1, 94.9, **56.5**) | (98.4, 53.0, 52.8) | (**99.0**, 92.2, 48.1) | (98.5, 95.3, 54.9) | (98.5, **96.2**, 53.0) |
| (AUROC, PRO, F1) | BTAD | 3 | (72.7, 27.5, 18.5) | (89.3, 68.7, 40.6) | (94.2, 75.4, 47.7) | (90.7, 22.3, 51.4) | (92.7, 70.0, 45.1) | (**96.4**, 78.9, 52.2) | (95.6, **86.1**, **52.7**) |
| | DAGM | 10 | (87.6, 65.7, 12.7) | (82.4, 66.0, 37.4) | (95.6, 91.0, 58.9) | (94.3, 42.5, 59.6) | (92.6, 79.7, 44.0) | (**98.2**, 95.0, 56.1) | (97.5, **95.2**, **61.2**) |
| | DTD-Synthetic | 12 | (79.5, 51.4, 16.1) | (95.2, 87.3, 67.4) | (97.9, 92.0, 62.2) | (98.5, 75.0, **71.8**) | (97.4, 88.9, 58.0) | (97.7, 92.3, 62.8) | (**99.0**, **96.5**, 67.5) |
| | AVERAGE | - | (79.9, 52.6, 16.0) | (90.9, 74.4, 42.2) | (95.6, 87.2, 47.0) | (94.7, 49.7, **49.9**) | (95.0, 83.8, 42.5) | (96.0, 89.1, 47.2) | (**96.2**, **92.3**, 49.6) |

Table 2: **ZSAD performance comparison on medical domain.** This table reports the performance (%) of different methods on seven medical datasets, evaluated at both image-level detection (AUROC, AP, F1) and pixel-level location (AUROC, PRO, F1). The best performance is shown in bold, with the second-best underlined. Note that image-level medical AD datasets do not provide segmentation annotations; therefore, the pixel-level and image-level datasets are distinct. Multi-seed results and standard deviations are provided in Appendix C.

| Task | Datasets | $|\mathcal{C}|$ | CLIP | WinCLIP | APRIL-GAN | AnomalyCLIP | AdaCLIP | AA-CLIP | FA-Prompt | ViP²-CLIP |
|---|---|---|---|---|---|---|---|---|---|---|
| | | | *OpenCLIP* | *CVPR 2023* | *CVPRw 2023* | *ICLR 2024* | *ECCV 2024* | *CVPR 2025* | *ICCV 2025* | *-* |
| Image-level | HeadCT | 1 | (67.8, 62.4, 70.9) | (81.8, 80.2, 78.9) | (89.1, 89.4, 82.1) | (93.0, 91.1, 88.4) | (94.0, 91.4, 90.1) | (88.1, 90.4, 80.6) | (**94.3**, 91.1, **90.5**) | (**94.3**, **93.9**, 88.1) |
| (AUROC, AP, F1) | BrainMRI | 1 | (72.2, 81.5, 76.5) | (86.6, 91.5, 84.1) | (89.4, 91.0, 88.2) | (90.0, 92.1, 86.5) | (94.3, 95.5, 92.2) | (92.2, 94.5, 88.6) | (95.1, 94.7, 92.1) | (**95.3**, **96.7**, **92.3**) |
| | Brain35H | 1 | (76.3, 77.7, 72.2) | (79.9, 82.2, 74.0) | (91.6, 92.1, 84.5) | (93.4, 93.8, 86.4) | (95.7, 95.8, 91.1) | (89.3, 90.6, 81.6) | (**97.6**, **97.1**, **91.4**) | (95.8, 96.0, 90.1) |
| | AVERAGE | - | (72.1, 73.9, 73.2) | (82.8, 84.6, 79.0) | (90.0, 90.8, 84.9) | (92.1, 92.3, 87.1) | (94.7, 94.2, 91.1) | (89.9, 91.8, 83.6) | (95.7, 94.3, 91.3) | (95.1, **95.5**, 90.2) |
| Pixel-level | ISIC | 1 | (43.2, 7.6, 44.0) | (83.3, 55.1, 64.1) | (89.4, 77.2, 71.4) | (89.4, 78.4, 71.6) | (91.3, 53.2, 75.5) | (**91.6**, **86.2**, **78.1**) | (87.8, 77.7, 69.2) | (90.3, 82.3, 73.7) |
| (AUROC, PRO, F1) | CVC-ColonDB | 1 | (59.8, 30.4, 17.8) | (64.8, 28.4, 21.0) | (78.4, 64.6, 29.7) | (81.9, 71.2, 37.5) | (81.5, 64.3, 33.6) | (80.6, 62.1, 32.3) | (**84.5**, 74.7, 40.2) | (82.5, **74.8**, 36.2) |
| | CVC-ClinicDB | 1 | (63.1, 33.8, 23.4) | (70.3, 32.5, 27.2) | (80.5, 60.7, 38.7) | (82.9, 68.1, 42.4) | (83.9, 65.7, 42.3) | (85.6, 66.3, 44.3) | (84.9, 70.8, **44.9**) | (**86.3**, **72.1**, **44.9**) |
| | Kvasir | 1 | (58.0, 18.3, 29.3) | (69.7, 24.5, 35.9) | (75.0, 36.3, 40.0) | (79.0, 45.4, 46.2) | (81.6, 49.1, 47.1) | (81.5, 48.4, **49.2**) | (80.0, 46.4, 47.3) | (**81.9**, 47.8, 47.3) |
| | AVERAGE | - | (56.0, 22.5, 28.6) | (72.0, 35.1, 37.1) | (80.8, 59.7, 45.0) | (83.3, 65.8, 49.4) | (84.6, 58.1, 49.6) | (84.8, 65.8, **51.0**) | (84.3, 67.4, 50.4) | (**85.3**, **69.3**, 50.5) |

training scheme, where the model is fine-tuned on one dataset and evaluated on another without any overlap. Specifically, we fine-tune ViP²-CLIP exclusively on the MVTec AD test split and evaluate its zero-shot performance on all other datasets. When evaluating on MVTec AD, we fine-tune the model on the VisA test set. We adopt a consistent evaluation protocol across all baselines. All experiments are implemented in PyTorch 2.6.0 and conducted on a single NVIDIA L20 GPU (48 GB). Further implementation details are provided in Appendix B.

## 5.2 Main Results

**Zero-Shot Anomaly Detection on Industrial Datasets** Table 1 compares ViP²-CLIP with six representative baselines across seven industrial defect benchmarks. Overall, our method achieves competitive performance, demonstrating improved mean performance across datasets. Early methods such as WinCLIP and APRIL-GAN primarily rely on handcrafted prompts and local feature tuning, but suffer from limited cross-modal interaction. AnomalyCLIP introduces object-agnostic prompts to improve generalization; however, its static prompt design limits fine-grained semantic alignment. AA-CLIP enhances discriminative capability by enlarging the margin in the text embedding space, while AdaCLIP further refines prompts in both visual and textual domains. However, despite achieving improved F1 scores, AdaCLIP overlooks the AUPRO metric, resulting in imbalanced performance in ZSAD. FAPrompt explores data-dependent abnormality priors, yet still struggles on industrial benchmarks. In contrast, ViP²-CLIP achieves a more balanced trade-off between detection and localization performance. At the pixel level, it surpasses FAPrompt by 3.2% in PRO and 2.4% in F1 on average, demonstrating strong segmentation capability across industrial benchmarks.

Furthermore, the robustness analysis in Appendix C.1 shows low run-to-run variation across different random seeds, indicating stable and reproducible performance. We further observe that the improvements on several industrial datasets, such as VisA, MPDD, and BTAD, generally exceed the reported standard deviations, suggesting relatively consistent gains over prior methods. On several remaining datasets, where the performance gaps remain within a narrow range, ViP$^2$-CLIP achieves statistically comparable results to recent competitive approaches. Overall, these results indicate that the proposed framework maintains competitive detection performance while providing strong and stable pixel-level localization capability across diverse industrial scenarios. Qualitative results in Figure 4 further show that ViP$^2$-CLIP produces anomaly maps with clearer and more consistent boundaries than existing methods.

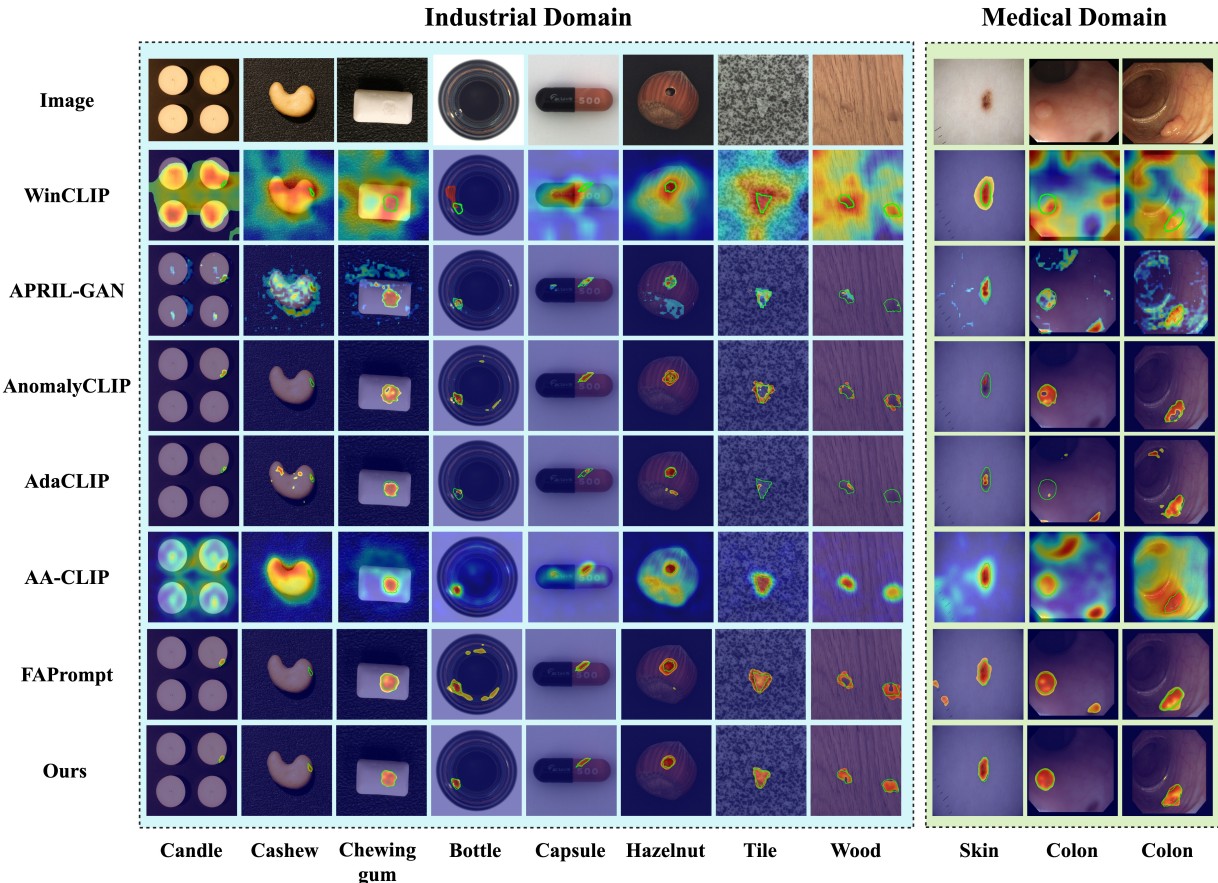

Figure 4: **Comparison of anomaly maps across ZSAD methods.** ViP$^2$-CLIP achieves sharper boundaries and more precise defect localization than competing methods.

**Zero-Shot Anomaly Detection on Medical Datasets** To evaluate cross-domain transferability, we further benchmark ViP$^2$-CLIP on seven medical anomaly detection datasets. As shown in Table 2, the proposed framework achieves competitive overall performance. In particular, ViP$^2$-CLIP obtains favorable results on datasets such as BrainMRI and CVC-ClinicDB, demonstrating the effectiveness of the proposed image-conditioned prompting mechanism in complex medical scenarios. On several remaining datasets, the performance gains are relatively limited, which we mainly attribute to the substantial domain gap between industrial auxiliary data and medical target domains. Specifically, the proposed ViP-Prompt facilitates semantic alignment by projecting global visual embeddings into the text prompt space through lightweight learnable adapters. When the auxiliary training data and target medical images exhibit significantly different feature distributions, the transferred visual-conditioned prompts may become less effective for accurate

Table 3: **Comparison of Computational Efficiency on VisA.** This table reports training time, inference latency, GPU memory usage, trainable parameters and encoder fine-tuning requirements across methods, all evaluated on the VisA dataset under a unified setup. The best results are shown in bold, with the second-best underlined.

| Model | Training Time (h) | Inference Time (ms) | GPU Cost (GB) | Trainable Parameters (M) | Encoder Fine-tuning |
|---|---|---|---|---|---|
| AnomalyCLIP | 1.02 | 90.72 ± 0.16 | 2.75 | 5.56 | ✓ |
| AdaCLIP | 2.24 | 134.20 ± 0.39 | 3.17 | 10.67 | ✓ |
| AA-CLIP | 0.93 | **19.17 ± 0.23** | 2.87 | 12.58 | ✓ |
| FAPrompt | 5.62 | 439.74 ± 0.024 | **2.07** | 9.61 | ✓ |
| ViP²-CLIP | **0.54** | 48.69 ± 0.19 | 2.21 | **4.15** | – |

Table 4: **Performance Comparison of Different Module Combinations.** This table reports the performance (%) of ViP-VCA, ViP-FGP, and UTPA on MVTec AD and VisA at both pixel and image levels. The full combination consistently achieves the best overall performance, demonstrating the complementary benefits of all modules.

| UTPA | ViP-Prompt | | MVTec AD | | VisA | |
|---|---|---|---|---|---|---|
| | VCA | FGP | Pixel-level | Image-level | Pixel-level | Image-level |
| | | | (37.8, 11.5, 7.0) | (74.1, 87.6, 87.2) | (43.5, 14.6, 2.8) | (60.2, 66.3, 74.5) |
| | ✓ | | (89.5, 79.9, 38.6) | (71.8, 86.2, 88.5) | (95.2, 90.9, 31.6) | (79.7, 82.8, 79.5) |
| | | ✓ | (90.0, 85.5, 42.7) | (66.1, 82.6, 86.6) | (95.3, 91.5, 33.4) | (60.1, 66.4, 73.2) |
| | ✓ | ✓ | (87.7, 83.8, 42.4) | (88.9, 94.9, 91.6) | (95.2, 91.6, **35.5**) | (83.0, 85.9, 81.4) |
| ✓ | ✓ | | (89.8, 84.4, 40.2) | (89.2, 95.0, 91.4) | (94.9, 91.4, 31.3) | (86.3, 89.1, 83.8) |
| ✓ | | ✓ | (88.8, 82.1, 38.7) | (85.2, 93.3, 89.0) | (95.3, 91.9, 31.4) | (87.6, 89.4, 83.8) |
| ✓ | ✓ | ✓ | (**90.5, 87.1, 43.1**) | (**91.2, 96.0, 92.0**) | (**95.4, 92.2**, 33.6) | (**88.5, 90.4, 84.8**) |

cross-domain semantic alignment. We provide additional analyses in Appendix C.4, where the results suggest that introducing medically related auxiliary data during finetuning can further improve prompt alignment quality and localization performance on medical images.

Nevertheless, the qualitative results presented in Figure 4 provide additional evidence of its localization capability. Specifically, ViP²-CLIP accurately localizes melanoma lesions in dermoscopic images and colonic polyps in endoscopic frames, highlighting its adaptability to diverse pathological patterns.

**Computational Efficiency Analysis** Table 3 compares the computational efficiency of ViP²-CLIP with four representative ZSAD methods on the VisA dataset. For consistency, all methods are evaluated under the same hardware environment, inference protocol, and input resolution. Since existing methods employ different optimization strategies, we follow their official implementations and default training configurations, as summarized in Appendix B.2. As shown in Table 3, the compared methods all require CLIP encoder fine-tuning, which incurs considerably higher training time, GPU memory consumption, and trainable parameter counts. In contrast, ViP²-CLIP achieves competitive performance without requiring encoder fine-tuning. By fully freezing the CLIP backbone and optimizing only two lightweight adapters, the proposed framework substantially reduces training overhead and computational cost, making it a practical and efficient solution for deployment in resource-constrained scenarios.

## 5.3 Ablation Study

**Module Ablation** We systematically evaluate the contributions of the three key components: ViP-VCA, ViP-FGP, and UTPA. As shown in Table 4, different module combinations exhibit distinct effects on image-level detection and pixel-level localization performance, highlighting the complementary roles of the proposed components. Specifically, ViP-VCA injects global visual context into the prompt embedding space through learnable parameters, improving semantic alignment for both image-level detection and pixel-level localization. In contrast, ViP-FGP incorporates fine-grained local visual cues that enhance sensitivity to subtle regional anomalies. However, when used alone, FGP also leads to noticeable drops in image-level AUROC on both MVTec AD and VisA datasets. This observation indicates that emphasizing local feature interactions

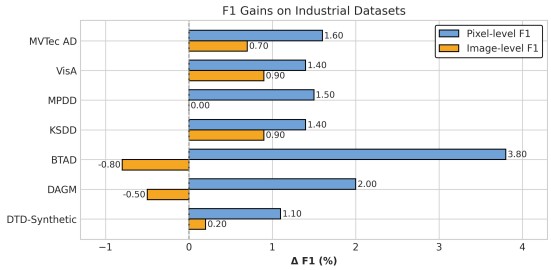

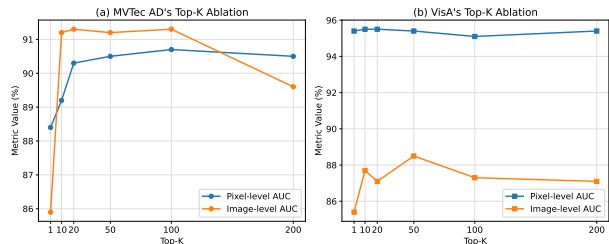

Figure 5: **F1 gains of visual-conditioned prompts over static prompts.** Consistent improvements across seven industrial datasets demonstrate robustness to domain shifts.

Figure 6: **Effect of Top-$K$ selection on anomaly detection.** AUROC on MVTec AD and VisA improves with larger $K$ and stabilizes around $K = 50$, balancing anomaly coverage and background noise.

Table 5: **Performance comparison between conventional dual-branch alignment and the proposed UTPA strategy.** Results are reported on MVTec AD and VisA under both controlled ablation settings and existing CLIP-based ZSAD frameworks. Replacing separate global-local alignment with UTPA improves the overall balance between image-level detection and pixel-level localization.

| Module | MVTec AD | | VisA | |
| --- | --- | --- | --- | --- |
| | Pixel-level | Image-level | Pixel-level | Image-level |
| CLIP_DUAL | (69.6, 24.6, 12.0) | (**88.6**, **94.9**, **91.3**) | (93.3, 82.4, 24.2) | (79.3, 82.4, 79.0) |
| CLIP_UTPA | (**89.6**, **83.7**, **37.4**) | (85.0, 93.3, 88.8) | (**94.4**, **88.4**, **26.2**) | (**83.8**, **86.7**, **81.4**) |
| AnomalyCLIP | (91.1, 81.4, 39.1) | (91.3, 96.4, 92.7) | (**95.5**, 86.7, 28.3) | (82.0, 85.3, 80.4) |
| AnomalyCLIP_UTPA | (**91.3**, **83.2**, **41.3**) | (**91.5**, **96.7**, **93.1**) | (**95.5**, **88.6**, **30.1**) | (**84.4**, **87.5**, **82.9**) |

improves sensitivity to subtle regional anomalies, while partially weakening global semantic consistency that benefits image-level anomaly scoring. When ViP-VCA and ViP-FGP are combined, the model achieves a better balance between global semantic understanding and local anomaly perception. Building upon this combination, UTPA further unifies the optimization objectives of detection and localization through consistent text-patch alignment strategy, resulting in more balanced image-level and pixel-level performance.

**Static Prompts vs. Visual-Perception Prompts** To quantify the benefits of visual conditioning, we compare ViP$^2$-CLIP against a baseline variant (ViP$^2$-CLIP$_{re}$) that relies solely on static learnable prompts without visual cues. Figure 5 illustrates the F1-score gains across the seven industrial datasets. ViP$^2$-CLIP consistently achieves higher pixel-level F1 on all datasets and improves image-level F1 on five of them. We attribute the slight image-level drops on BTAD and DAGM to significant domain shifts relative to the MVTec AD training data. Specifically, BTAD features real-world industrial settings with complex lighting, while DAGM contains synthetic textures with small, repetitive anomalies. Such domain shifts may affect the extracted global visual embeddings, thereby reducing the transferability of the ViP-VCA module. Despite these minor fluctuations, ViP-Prompt maintains strong performance across diverse scenarios. Rather than explicitly improving robustness to class-name variations, it replaces fixed class tokens with image-conditioned prompts, thereby removing reliance on category labels and reducing sensitivity to wording differences. This design effectively mitigates segmentation instability under ambiguous or privacy-constrained annotations, leading to improved generalization robustness.

**Top-$K$ Selection Ablation** To evaluate the impact of the number of anomalous patches on the image-level representation, we vary $K \in \{1, 10, 20, 50, 100, 200\}$ and report the AUROC in Figure 6. In UTPA, Top-$K$ pooling aggregates the $K$ most anomalous patches to compute the image-level score. A small $K$ may miss spatially distributed defects, whereas a large $K$ risks diluting the anomaly signals with normal background patches. As illustrated in Figure 6, performance steadily improves as $K$ increases, stabilizing around $K = 50$ on both MVTec AD and VisA (reaching the optimum on VisA and near-optimum on MVTec AD, with

Table 6: **Performance Comparison of Different State Adjectives in Prompts.** This table reports the performance (%) of different adjective pairs used in prompt construction, evaluated on MVTec AD and VisA at both pixel and image levels. The results demonstrate consistent performance across variants, indicating robustness to specific adjective choices.

| State words | MVTec AD | | VisA | |
|---|---|---|---|---|
| | Pixel-level | Image-level | Pixel-level | Image-level |
| Good/Damage | (90.5, 87.1, 43.1) | (91.2, 96.0, 92.0) | (95.4, 92.2, 33.6) | (88.5, 90.4, 84.8) |
| Normal/Abnormal | (90.6, 87.5, 43.6) | (91.1, 95.9, 92.3) | (95.3, 91.9, 33.3) | (88.3, 90.2, 84.5) |
| Perfect/Flawed | (90.4, 87.2, 43.7) | (91.7, 96.2, 92.5) | (95.4, 92.0, 33.5) | (88.3, 90.3, 84.5) |
| Flawless/Imperfect | (90.4, 87.4, 43.3) | (91.2, 95.9, 92.2) | (95.4, 91.9, 33.1) | (87.9, 90.2, 84.3) |

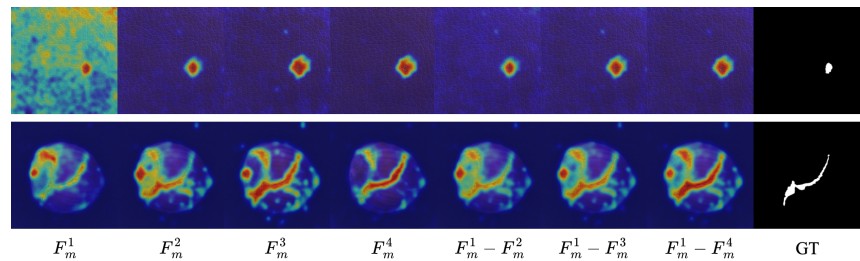

$F_m^1$    $F_m^2$    $F_m^3$    $F_m^4$    $F_m^1 - F_m^2$    $F_m^1 - F_m^3$    $F_m^1 - F_m^4$    GT

Figure 7: **Visualization of anomaly maps across different feature layers.** Deeper layers improve localization for complex objects, while intermediate layers suffice for simpler textures. This highlights the complementary nature of multi-level features for robust anomaly detection.

consistent behavior observed between $K = 50$ and $100$). Consequently, we set $K = 50$ as the default value to extract image-level descriptors, avoiding per-dataset tuning while ensuring robust anomaly characterization.

**UTPA for Mitigating Optimization Conflicts**   We conduct controlled ablation studies to evaluate the effectiveness of UTPA in mitigating the optimization conflict between image-level detection and pixel-level localization. Specifically, we first compare two alignment strategies using a frozen CLIP backbone with learnable normal and anomalous prompts. CLIP_DUAL adopts the conventional dual-branch design that independently aligns prompts with global and local visual features, whereas CLIP_UTPA replaces this formulation with the proposed unified text-patch alignment strategy. As shown in Table 5, CLIP_DUAL tends to favor image-level optimization while exhibiting relatively weak localization performance. In contrast, CLIP_UTPA achieves substantially better pixel-level results while maintaining competitive image-level performance, indicating that unified local alignment provides a more balanced optimization objective. To further isolate the effect of the alignment strategy, we additionally replace the original dual-branch alignment in AnomalyCLIP with UTPA while keeping all other components unchanged. The resulting variant, AnomalyCLIP_UTPA, achieves notable improvements in pixel-level localization on MVTec AD and in both image-level detection and pixel-level localization on VisA, demonstrating that the proposed alignment strategy can be effectively integrated into existing CLIP-based ZSAD frameworks. These results together suggest that UTPA helps improve optimization by concentrating the alignment objective on local visual embeddings and text embeddings, providing more consistent training signals for anomaly detection and localization.

**State-Adjective Ablation**   To verify the robustness of our prompt design, we replace the original 'good/damaged' adjectives with several alternative pairs of similar meanings and retrain the model on both MVTec-AD and VisA datasets. As Table 6 demonstrates, detection performance remains relatively consistent across all variants, indicating insensitivity to specific adjective choices. This resilience stems from our learnable prompt tokens, which autonomously capture normal and anomalous semantics without relying on rigid handcrafted templates.

**Visualization at Different Layers**   To analyze the impact of model depth on anomaly representations used in UTPA, Figure 7 presents anomaly maps derived from layers $F_m^1$ through $F_m^4$. For objects with complex structures, such as hazelnuts, deeper layers yield more accurate anomaly localization. Conversely, for objects with simpler textures, such as leather, intermediate layers provide sufficient discriminative power. By aggregating features across multiple depths, our model integrates complementary contextual cues, enabling it to robustly adapt to diverse anomaly types.

# 6   Conclusions

We propose ViP$^2$-CLIP, a lightweight CLIP-based ZSAD framework for detecting anomalies in unseen categories without access to target-domain training samples, relying solely on auxiliary data. At its core, the ViP-Prompt module adaptively integrates global and local visual cues into learnable text prompts, reducing the reliance on manual templates and explicit class-name priors, and facilitating more stable cross-modal alignment. Furthermore, the UTPA strategy enforces consistent text–patch alignment across multiple scales, helping alleviate task conflicts and improving both detection and localization performance. Extensive experiments across 14 benchmarks demonstrate the competitive performance and efficiency of ViP$^2$-CLIP. Overall, the proposed framework provides a practical solution for zero-shot anomaly detection, particularly in scenarios with ambiguous or unavailable category labels.

**Ethical and Deployment Considerations**   This work uses only publicly available datasets and follows their respective licenses and usage protocols. All industrial and medical benchmarks employed in our experiments are widely adopted research datasets that have been anonymized and released for academic purposes. Although ViP$^2$-CLIP demonstrates promising zero-shot anomaly localization capability on several medical benchmarks, the proposed framework is intended for research purposes and should not be considered a standalone diagnostic system.

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

# A   Motivation Statement

## A.1   Robustness Analysis of Class Names in Prompts

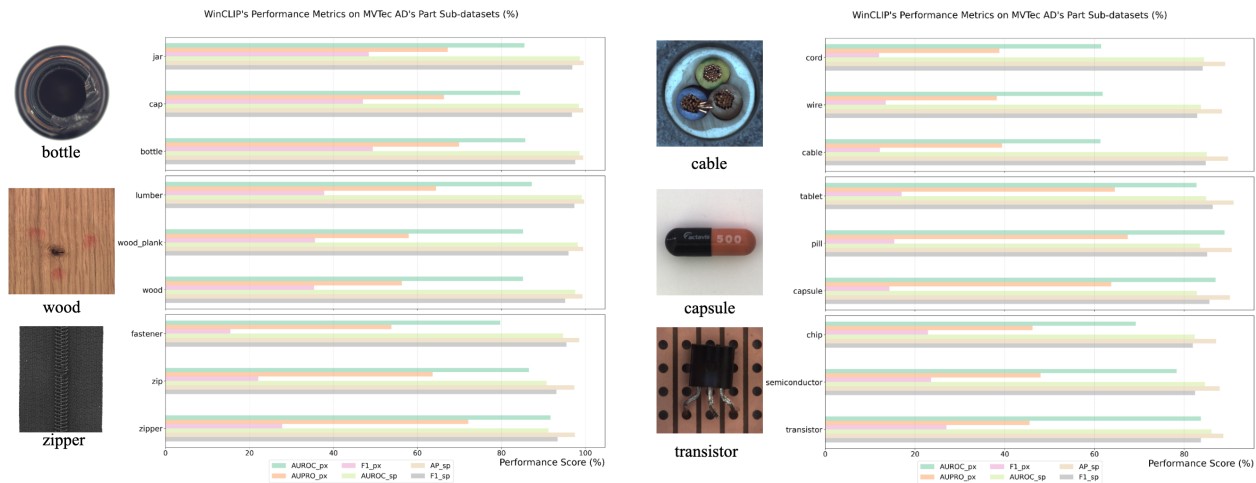

Figure 8: WinCLIP's performance under different class-name variations.

Figure 9: WinCLIP's performance under different class-name variations.

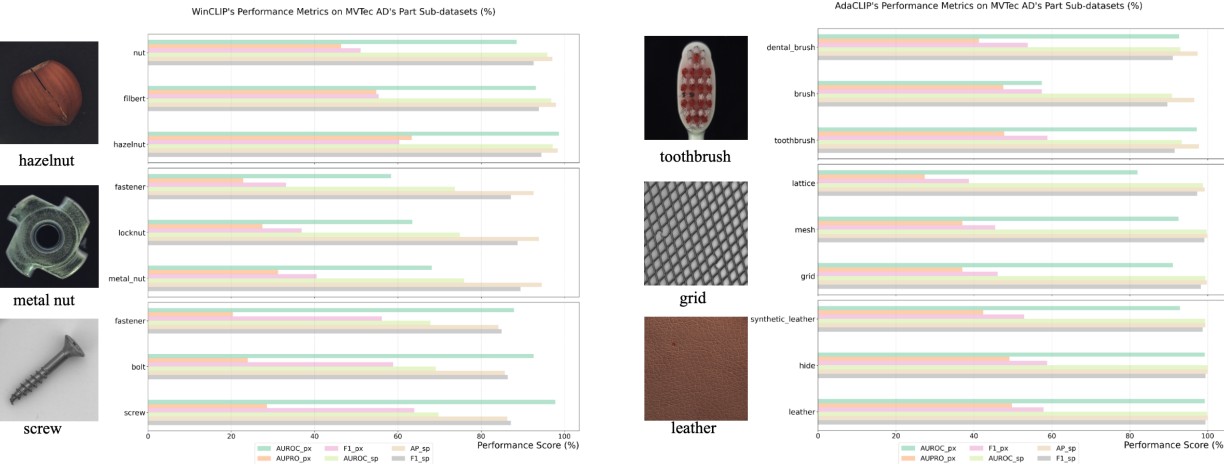

Figure 10: WinCLIP's performance under different class-name variations.

Figure 11: AdaCLIP's performance under different class-name variations.

Methods such as WinCLIP (Jeong et al., 2023), CLIP-AD (Chen et al., 2024), AdaCLIP (Cao et al., 2024), and KANOCLIP (Li et al., 2025) explicitly incorporate class names into their prompting pipelines to enhance semantic discrimination. However, this design also introduces sensitivity to class-name variations, which can negatively affect anomaly localization.

To systematically evaluate this effect, we conduct synonym-substitution experiments across all 15 categories of the MVTec AD dataset using both the training-free WinCLIP model and the finetuned AdaCLIP model. For each category, we replace the original class name with three semantically equivalent variants (e.g., replacing "zipper" with "zip" or "fastener") and report the mean and standard deviation across all substitutions.

The qualitative results in Figure 8-Figure 12 and the quantitative summary in Table 7 reveal a consistent trend across both methods: image-level classification metrics remain relatively stable under synonym substitutions (std typically ≤

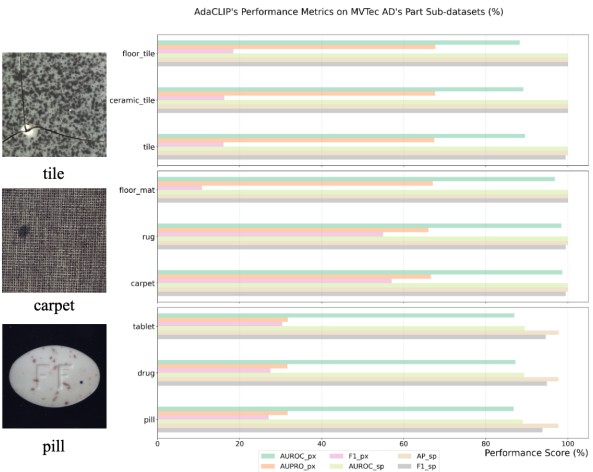

Figure 12: AdaCLIP's performance under different class-name variations.

1.0), whereas pixel-level localization metrics exhibit substantially larger fluctuations. For instance, in AdaCLIP, the pixel-level F1 on the *carpet* category varies by up to 26.1 points, and the AUPRO on *toothbrush* fluctuates by 21.8 points. Similar instability is observed in categories with subtle or localized defects, such as *zipper*, *transistor*, and *screw*. Notably, although AdaCLIP improves overall performance through auxiliary fine-tuning, this sensitivity to class-name perturbations is not fully mitigated. This observation further motivates the need for label-agnostic prompt designs, especially in industrial scenarios with ambiguous category definitions or privacy-constrained settings where explicit class labels may be unavailable.

Table 7: Sensitivity analysis under synonym substitutions across all 15 MVTec AD categories. Values are reported as mean ± standard deviation over three synonym variants per category. Results marked with * are obtained using the training-free WinCLIP model, while the remaining categories are evaluated using the finetuned AdaCLIP model.

| Category | Pixel-level | | | Image-level | | |
|---|---|---|---|---|---|---|
| | AUROC | AUPRO | F1 | AUROC | AP | F1 |
| bottle* | 85.2 ± 0.7 | 67.8 ± 1.9 | 48.3 ± 1.2 | 98.6 ± 0.2 | 99.5 ± 0.1 | 97.1 ± 0.4 |
| wood* | 85.8 ± 1.2 | 59.5 ± 4.3 | 36.3 ± 1.3 | 98.3 ± 0.8 | 99.5 ± 0.2 | 96.2 ± 1.1 |
| zipper* | 86.0 ± 6.0 | 63.2 ± 9.2 | 21.8 ± 6.2 | 92.2 ± 2.2 | 97.8 ± 0.6 | 94.0 ± 1.3 |
| cable* | 61.5 ± 0.3 | 38.8 ± 0.6 | 12.6 ± 0.8 | 84.4 ± 0.7 | 89.1 ± 0.7 | 83.9 ± 1.0 |
| capsule* | 86.2 ± 3.2 | 65.2 ± 1.9 | 15.6 ± 1.4 | 83.7 ± 1.1 | 90.6 ± 0.4 | 85.7 ± 0.6 |
| transistor* | 77.1 ± 7.3 | 46.6 ± 1.3 | 24.5 ± 2.2 | 84.3 ± 1.9 | 87.9 ± 0.8 | 82.7 ± 0.9 |
| hazelnut* | 93.4 ± 5.1 | 54.8 ± 8.5 | 55.6 ± 4.6 | 96.7 ± 0.7 | 97.8 ± 0.6 | 93.6 ± 0.9 |
| metal_nut* | 63.3 ± 4.9 | 27.2 ± 4.2 | 36.9 ± 3.7 | 74.8 ± 1.1 | 93.7 ± 1.0 | 88.4 ± 1.2 |
| screw* | 92.8 ± 5.0 | 24.3 ± 4.1 | 59.6 ± 4.0 | 68.9 ± 1.0 | 85.4 ± 1.1 | 86.1 ± 1.1 |
| toothbrush | 82.4 ± 21.8 | 45.5 ± 3.7 | 56.7 ± 2.6 | 92.4 ± 1.4 | 97.2 ± 0.6 | 90.8 ± 1.0 |
| grid | 88.5 ± 5.7 | 33.8 ± 5.6 | 43.4 ± 4.1 | 99.3 ± 0.5 | 99.6 ± 0.4 | 98.2 ± 0.9 |
| leather | 97.1 ± 3.7 | 47.1 ± 4.1 | 56.5 ± 3.2 | 99.7 ± 0.3 | 99.8 ± 0.3 | 99.2 ± 0.4 |
| tile | 89.0 ± 0.7 | 67.6 ± 0.1 | 16.9 ± 1.3 | 100.0 ± 0.0 | 100.0 ± 0.0 | 99.8 ± 0.3 |
| carpet | 97.9 ± 1.0 | 66.6 ± 0.5 | 40.9 ± 26.1 | 100.0 ± 0.0 | 100.0 ± 0.0 | 99.6 ± 0.3 |
| pill | 86.9 ± 0.2 | 31.7 ± 0.1 | 28.3 ± 1.8 | 89.2 ± 0.3 | 97.7 ± 0.0 | 94.4 ± 0.6 |

## A.2 Comparative Analysis under Ambiguous Category Labels

To further quantify the practical gap between class-name-dependent methods and the proposed label-agnostic ViP²-CLIP framework under ambiguous category descriptions, following the above motivation analysis, we conduct an additional comparative analysis on three representative MVTec AD categories: *bottle*, *zipper*, and *carpet*. Specifically, we evaluate both the training-free WinCLIP (Jeong et al., 2023) and the finetuned AdaCLIP (Cao et al., 2024) under multiple semantically similar category-name substitutions, including (*bottle*, *cap*, *jar*), (*zipper*, *zip*, *fastener*), and (*carpet*, *rug*, *floor_mat*). These substitutions preserve highly similar semantics while simulating realistic industrial scenarios where category descriptions may be ambiguous, inconsistent, or unavailable.

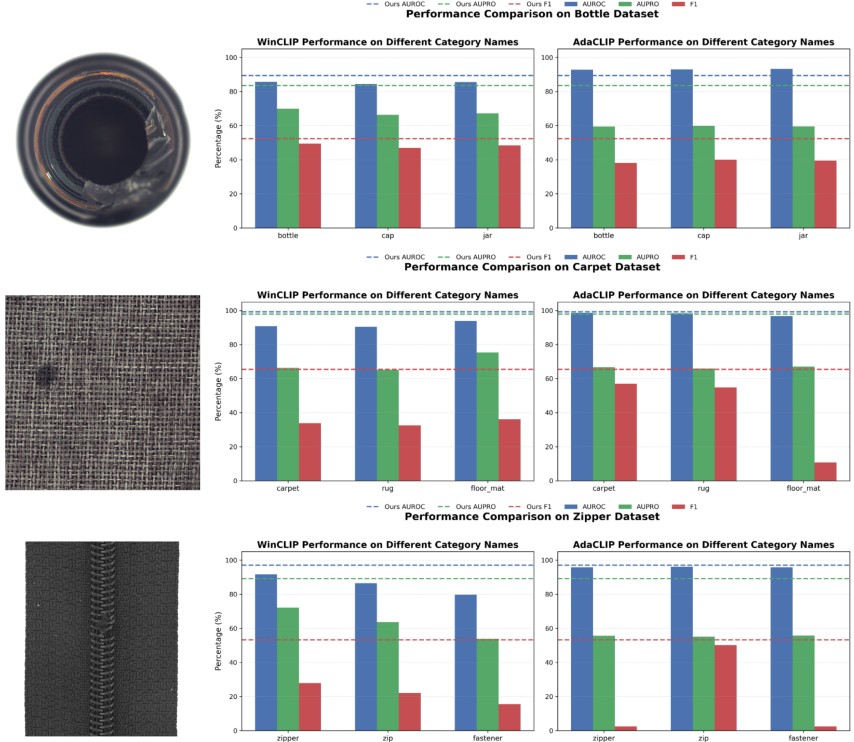

Figure 13: Comparative analysis under ambiguous category descriptions. Performance comparison of WinCLIP, AdaCLIP, and ViP²-CLIP under semantically equivalent category-name substitutions on the *bottle*, *zipper*, and *carpet* categories of MVTec AD.

As illustrated in Fig. 13, both WinCLIP and AdaCLIP exhibit substantial performance fluctuations under different category-name substitutions. In particular, in the *zipper* category, AdaCLIP achieves relatively strong performance when using the synonym "zip", but its localization performance deteriorates substantially when using "zipper" or "fastener", resulting in extremely low pixel-level F1 scores and blurred anomaly boundaries. In contrast, ViP²-CLIP maintains comparatively stable and competitive localization performance in terms of PRO and F1 across varying category descriptions, demonstrating stronger robustness against category ambiguity. These results further suggest that the proposed image-conditioned prompting mechanism facilitates more reliable semantic alignment without relying on manually specified class names.

Overall, the experimental results consistently demonstrate that ViP²-CLIP provides a more robust and reliable localization baseline under category ambiguity, making the framework particularly suitable for practical industrial scenarios involving ambiguous, inconsistent, or privacy-constrained category descriptions.

# B    Experimental Details

## B.1    Implementation Details

We employ the publicly released CLIP (ViT-L/14@336px) model as our frozen feature extractor. Across all experiments, the learnable prompt length is fixed at 10 tokens, 3 of which are dynamically fused with the global visual feature. For the UTPA strategy, we set Top-$K = 50$ and enforce cross-modal alignment at intermediate layers $\{6, 12, 18, 24\}$. During training, input images are resized to $518 \times 518$. We optimize the model using the Adam optimizer with a learning rate of $1 \times 10^{-3}$ and a batch size of 8, training for a total of 10 epochs. All experiments are implemented in PyTorch 2.6.0 and executed on a single NVIDIA L20 GPU (48 GB).

We conduct comprehensive evaluations across 14 public benchmarks, covering both industrial and medical domains. Following the zero-shot protocol, we fine-tune our model exclusively on the MVTec AD test set and perform zero-shot evaluation on all other datasets. Conversely, when evaluating on MVTec AD, the model is fine-tuned solely on the VisA test set. For evaluation metrics, we report the Area Under the ROC Curve (AUROC), Average Precision (AP), and F1-score (F1) for image-level anomaly detection. For pixel-level segmentation, we report AUROC, Per-Region Overlap (PRO), and F1-score.

## B.2    Baselines

To validate the superiority of ViP$^2$-CLIP, we benchmark it against the following state-of-the-art ZSAD methods:

- **WinCLIP** (Jeong et al., 2023): The pioneering CLIP-based ZSAD framework. It utilizes an ensemble of manually designed prompts alongside a multi-window feature extraction strategy. We evaluate this method using its official parameters.

- **APRIL-GAN** (Chen et al., 2023): An extension of WinCLIP that incorporates learnable linear projection layers to enrich local visual representations. All experiments are conducted using the official weights, strictly adhering to the original implementation protocols.

- **AnomalyCLIP** (Zhou et al., 2024): This method introduces object-agnostic learnable prompts and a DPAM mechanism to strengthen local feature modeling. Evaluations are performed using the publicly released official model weights.

- **AdaCLIP** (Cao et al., 2024): AdaCLIP integrates hybrid prompts into both the text and image encoders, utilizing an HSF module for improved cross-modal fusion. Because its original evaluation protocol differs from ours (training on both MVTec AD and CVC-ColonDB before zero-shot testing), we reimplement AdaCLIP under our unified protocol to ensure a fair comparison, keeping all architectural parameters identical to the original paper.

- **AA-CLIP** (Ma et al., 2025a): This approach adopts a two-stage training strategy to enlarge the semantic gap between normal and anomalous texts via joint optimization of the text and image encoders. We reimplement AA-CLIP under our unified evaluation protocol, maintaining all original hyperparameter settings.

- **FAPrompt** (Zhu et al., 2025): FAPrompt advances ZSAD by learning fine-grained abnormality prompts to capture diverse defect patterns. It employs a Compound Abnormality Prompt (CAP) module to decompose abnormality cues, and a Data-dependent Abnormality Prior (DAP) module to dynamically adapt prompts to each test image. We carefully reimplement FAPrompt under our unified protocol, strictly preserving the original design and official hyperparameter settings.

Table 8:    Training configurations used in the computational efficiency comparison.

| Method | Image Size | Batch Size | Epochs | Precision | Encoder Fine-tuning |
|---|---|---|---|---|---|
| AnomalyCLIP | 518 | 8 | 15 | FP32 | ✓ |
| AdaCLIP | 518 | 1 | 5 | FP32 | ✓ |
| AA-CLIP | 518 | 2 | 25 | FP32 | ✓ |
| FAPrompt | 518 | 8 | 15 | FP32 | ✓ |
| ViP$^2$-CLIP | 518 | 8 | 10 | FP32 | – |

Table 9: Statistics of the datasets.

| Domain | Dataset | Category | Modalities | $|\mathcal{C}|$ | Normal and anomalous samples |
|---|---|---|---|---|---|
| Industrial | MVTec AD | Obj & texture | Photography | 15 | (467, 1258) |
| | VisA | | Photography | 12 | (962, 1200) |
| | MPDD | Obj | Photography | 6 | (176, 282) |
| | BTAD | | Photography | 3 | (451, 290) |
| | KSDD | | Photography | 1 | (286, 54) |
| | DAGM | Texture | Photography | 10 | (6996, 1054) |
| | DTD-Synthetic | | Photography | 12 | (357, 947) |
| Medical | ISIC | Skin | Photography | 1 | (0, 379) |
| | CVC-ClinicDB | | Endoscopy | 1 | (0, 612) |
| | CVC-ColonDB | Colon | Endoscopy | 1 | (0, 380) |
| | Kvasir | | Endoscopy | 1 | (0, 1000) |
| | HeadCT | | Radiology (CT) | 1 | (100, 100) |
| | BrainMRI | Brain | Radiology (MRI) | 1 | (98, 155) |
| | Br35H | | Radiology (MRI) | 1 | (1500, 1500) |

Table 10: Multi-seed evaluation of ViP$^2$-CLIP on industrial benchmarks. Results are reported as mean ± standard deviation over three random seeds, including image-level detection (AUROC, AP, F1) and pixel-level localization (AUROC, PRO, F1).

| Dataset | Image-level (AUROC, AP, F1) | Pixel-level (AUROC, PRO, F1) |
|---|---|---|
| MVTec AD | (91.5 ± 0.18, 96.2 ± 0.21, 92.1 ± 0.16) | (90.7 ± 0.19, 87.0 ± 0.13, 43.2 ± 0.37) |
| VisA | (88.5 ± 0.11, 90.3 ± 0.08, 84.9 ± 0.13) | (95.5 ± 0.08, 92.3 ± 0.31, 34.0 ± 0.21) |
| MPDD | (79.7 ± 0.21, 84.4 ± 0.27, 82.5 ± 0.19) | (97.1 ± 0.03, 92.5 ± 0.41, 35.8 ± 0.24) |
| KSDD | (98.1 ± 0.09, 95.8 ± 0.28, 93.1 ± 0.18) | (98.5 ± 0.18, 96.1 ± 0.17, 53.1 ± 0.19) |
| BTAD | (95.1 ± 0.26, 98.3 ± 0.19, 94.6 ± 0.17) | (95.9 ± 0.28, 86.0 ± 0.46, 52.5 ± 0.22) |
| DAGM | (98.5 ± 0.01, 94.4 ± 0.27, 92.6 ± 0.11) | (97.3 ± 0.04, 95.4 ± 0.06, 61.4 ± 0.37) |
| DTD-Synthetic | (95.6 ± 0.22, 98.1 ± 0.11, 94.4 ± 0.28) | (99.2 ± 0.19, 96.3 ± 0.38, 67.6 ± 0.31) |
| AVERAGE | (92.4 ± 0.16, 93.9 ± 0.23, 90.6 ± 0.19) | (96.4 ± 0.11, 92.2 ± 0.30, 49.7 ± 0.28) |

**Training Configurations for Efficiency Comparison** To improve the transparency of the efficiency comparison, we summarize the training configurations used by different methods in Table 8. All methods are evaluated under the same hardware environment, inference protocol, and input resolution. Since existing ZSAD methods employ different optimization strategies and training settings, we follow the official implementations and default configurations reported in their original papers. As shown in Table 8, most prior methods require partial or full encoder fine-tuning, resulting in increased optimization overhead during training. In contrast, ViP$^2$-CLIP keeps the CLIP backbone fully frozen and only optimizes two lightweight adapters, reducing training complexity and computational cost while maintaining competitive performance.

## B.3 Datasets

We evaluate our framework on 7 industrial datasets (MVTec AD, VisA, MPDD, BTAD, KSDD, DAGM, DTD-Synthetic) and 7 medical datasets (HeadCT, BrainMRI, Br35H, ISIC, CVC-ClinicDB, CVC-ColonDB, Kvasir), totaling 14 benchmarks. All methods are trained and evaluated strictly on each dataset's test split. Detailed dataset statistics are provided in Table 9. We apply OpenCLIP's default normalization to all input images and resize them to $(518, 518)$ to ensure appropriately scaled feature maps.

Table 11: Multi-seed evaluation of ViP²-CLIP on medical benchmarks. Results are reported as mean ± standard deviation over three random seeds, including image-level detection (AUROC, AP, F1) and pixel-level localization (AUROC, PRO, F1). Note that the image-level and pixel-level medical datasets are distinct due to the absence of segmentation annotations in the former.

| Dataset | Image-level (AUROC, AP, F1) | Pixel-level (AUROC, PRO, F1) |
|---|---|---|
| HeadCT | (94.3 ± 0.11, 94.0 ± 0.02, 88.6 ± 0.28) | - |
| BrainMRI | (95.5 ± 0.10, 96.5 ± 0.26, 92.4 ± 0.12) | - |
| Brain35H | (96.1 ± 0.08, 96.2 ± 0.17, 90.4 ± 0.26) | - |
| ISIC | - | (90.2 ± 0.02, 82.7 ± 0.22, 74.1 ± 0.31) |
| CVC-ColonDB | - | (82.9 ± 0.15, 74.9 ± 0.24, 36.6 ± 0.26) |
| CVC-ClinicDB | - | (86.5 ± 0.13, 71.9 ± 0.18, 45.1 ± 0.17) |
| Kvasir | - | (82.1 ± 0.19, 48.0 ± 0.21, 47.5 ± 0.20) |
| AVERAGE | (95.3 ± 0.10, 95.6 ± 0.21, 90.5 ± 0.26) | (85.4 ± 0.13, 69.4 ± 0.20, 50.8 ± 0.23) |

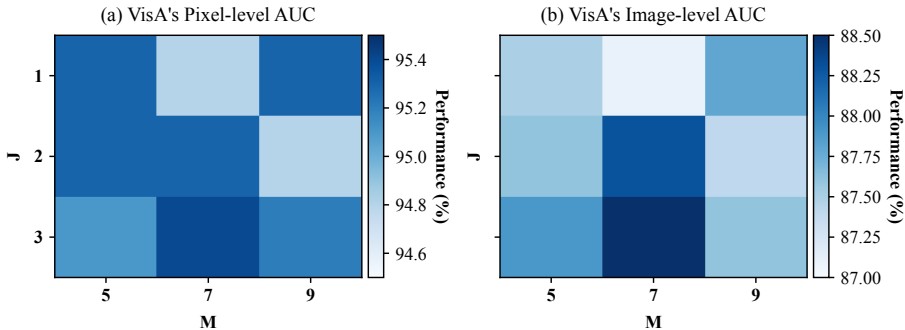

Figure 14: Prompt Length ablation.

# C Additional Experiments and Analysis

## C.1 Robustness and Stability Analysis

To evaluate the robustness of ViP²-CLIP under different random initializations, we conduct multi-seed experiments using three random seeds. Table 10 and Table 11 report the results on both industrial and medical benchmarks. Across all averaged metrics, the standard deviations remain consistently below 0.3%, indicating stable training and inference behavior. Importantly, the observed variations are substantially smaller than the performance gains reported in the main tables, suggesting that the improvements are not due to random fluctuations but are consistently brought by our method.

## C.2 Prompt Length Ablation

Based on VisA dataset, we vary the total prompt length to assess its effect on ZSAD performance, focusing on both the number of static learnable tokens $M$ and dynamic tokens $J$, which fuse with global visual context. Fig. 14 shows that increasing length does not always improve performance: too many learnable tokens introduce redundancy and risk overfitting, harming precision and generalization capacity. To balance between semantic expressiveness and model robustness, we set the prompt length to 10 by default, comprising 3 dynamic tokens infused with global visual semantics and 7 static tokens modelling generic normal and anomalous patterns.

## C.3 Comparison with Full-shot Methods

In this section, we compare ViP²-CLIP against two state-of-the-art full-shot methods, PatchCore (Roth et al., 2022) and RD4AD (Deng and Li, 2022), on five industrial datasets for which training samples are available. Results are presented in Table 12. Despite not using any target domain training samples, ViP²-CLIP achieves detection and

segmentation performance comparable to these fully supervised approaches, with particularly strong gains on the BTAD and DAGM datasets. This demonstrates that ViP$^2$-CLIP's visual-perception prompt mechanism adaptively generates fine-grained normal and anomalous text descriptions that effectively enhance CLIP's ZSAD capability, reducing the dependency on in-domain training data while maintaining strong generalization capability.

Table 12: Comparison of ZSAD performance between ViP$^2$-CLIP and SOTA full-shot methods. Results of methods with $^*$ are copied from the AnomalyCLIP paper. The best performance is highlighted in red, and the second is highlighted in blue.

| Task | Category | Datasets | PatchCore* | RD4AD* | ViP$^2$-CLIP |
|---|---|---|---|---|---|
| Image-level (AUROC, AP) | Obj &texture | MVTec AD | (**99.0**, **99.7**) | (_98.7_, _99.4_) | (91.2, 96.0) |
| | | VisA | (_94.6_, **95.9**) | (**95.3**, _95.7_) | (88.5, 90.4) |
| | Obj | MPDD | (**94.1**, **96.3**) | (_91.6_, _93.8_) | (79.7, 84.5) |
| | | BTAD | (93.2, **98.6**) | (_93.8_, 96.8) | (**95.0**, _98.4_) |
| | Texture | DAGM | (92.7, _81.3_) | (_92.9_, 79.1) | (**98.5**, **94.3**) |
| Pixel-level (AUROC, PRO) | Obj &texture | MVTec AD | (**98.1**, _92.8_) | (_97.8_, **93.6**) | (90.5, 87.1) |
| | | VisA | (**98.5**, **92.2**) | (_98.4_, 91.2) | (95.4, _92.2_) |
| | Obj | MPDD | (**98.8**, _94.9_) | (_98.4_, **95.2**) | (97.2, 92.6) |
| | | BTAD | (_97.4_, 74.4) | (**97.5**, _75.1_) | (95.6, **86.1**) |
| | Texture | DAGM | (95.9, 87.9) | (_96.8_, _91.9_) | (**97.5**, **95.2**) |

## C.4 Targeted Fine-tuning on Medical Data

Although ViP$^2$-CLIP achieves strong results on industrial benchmarks, its performance on medical images is relatively weaker, partly due to the domain gap between industrial data and medical target domains. To investigate whether fine-tuning on medical auxiliary data can bridge this gap, we design a cross-domain fine-tuning protocol: we train on MVTec AD and CVC-ColonDB test split and test on the remaining medical datasets; for CVC-ColonDB, we use VisA and CVC-ClinicDB test sets as auxiliary data. We compare against AnomalyCLIP under the same training strategy. As shown in Table 13, introducing medical auxiliary data significantly boosts detection performance, particularly on CVC-ClinicDB and Endo datasets, highlighting the critical role of auxiliary data in cross-domain generalization. Overall, ViP$^2$-CLIP achieves stronger overall performance than AnomalyCLIP under medical fine-tuning, demonstrating effective cross-domain anomaly detection capability.

Compared with Table 2 in the main text, we further analyze the impact of domain shift on model learning. Specifically, industrial and medical domains exhibit fundamentally different feature distributions and anomaly characteristics. Industrial anomalies are typically associated with repetitive textures, structural defects, or clear appearance corruptions under relatively clean and well-illuminated imaging conditions. In contrast, medical anomalies often appear in low-contrast regions with ambiguous boundaries and large intra-class variations, making abnormal patterns substantially

Table 13: ZSAD performance on medical images after fine-tuning by medical-domain datasets. Best performance is shown in red.

| Task | Datasets | $|\mathcal{C}|$ | AnomalyCLIP | ViP$^2$-CLIP |
|---|---|---|---|---|
| Image-level (AUROC, AP, F1) | HeadCT | 1 | (94.3, 94.0, 90.5) | (**96.5**, **96.5**, **92.8**) |
| | BrainMRI | 1 | (94.7, 95.7, 92.5) | (**96.7**, **97.6**, **94.9**) |
| | Brain35H | 1 | (96.5, 96.7, 92.6) | (**97.1**, **97.5**, **93.4**) |
| | AVERAGE | - | (95.2, 95.5, 91.9) | (**96.8**, **97.2**, **93.7**) |
| Pixel-level (AUROC, PRO, F1) | ISIC | 1 | (87.8, 75.6, 69.7) | (**91.7**, **82.9**, **75.3**) |
| | CVC-ColonDB | 1 | (**87.8**, 76.9, 48.5) | (87.7, **86.2**, **53.5**) |
| | CVC-ClinicDB | 1 | (88.8, 76.1, 51.7) | (**92.2**, **84.3**, **60.8**) |
| | Endo | 1 | (89.0, 73.5, 58.1) | (**92.3**, **82.8**, **65.2**) |
| | Kvasir | 1 | (85.6, 50.0, 55.1) | (**90.5**, **56.9**, **63.4**) |
| | AVERAGE | - | (87.8, 70.4, 56.6) | (**90.9**, **78.6**, **63.6**) |

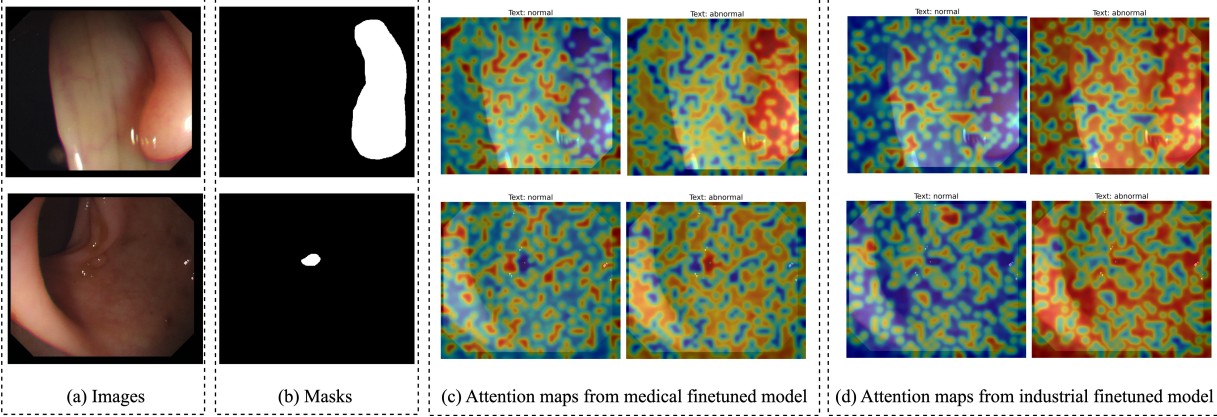

Figure 15: Attention visualization under different fine-tuning domains. Comparison of normal and anomalous prompt attention maps on CVC-ClinicDB using models trained with industrial-only auxiliary data and models additionally finetuned with medical auxiliary data.

more difficult to distinguish from surrounding tissues. Such discrepancies make CLIP representations learned from industrial textures and shapes difficult to directly align with medical lesion semantics.

To better illustrate this effect, we visualize the attention maps of both normal and anomalous prompts on the CVC-ClinicDB dataset under two finetuning settings: (1) training with industrial auxiliary data only (MVTec AD), and (2) additional fine-tuning with medical auxiliary data (CVC-ColonDB). The qualitative comparisons are shown in Fig. 15. When trained only on industrial data, the abnormal prompts exhibit spatially diffuse and structurally ambiguous attention responses on medical images, often failing to accurately focus on lesion regions. As a result, the finetune learning becomes less discriminative, leading to degraded lesion localization and detection performance. In contrast, after introducing medical auxiliary data during finetuning, the attention maps become substantially more concentrated and better aligned with lesion regions. These results suggest that medically related auxiliary data helps bridge the representation gap between industrial and medical domains, thereby improving prompt attention quality and cross-domain anomaly detection performance.

## C.5  Failure Cases and Future Work

While ViP$^2$-CLIP excels in clean, controlled environments with localized anomalies, it encounters challenges in more complex scenarios. We identify three primary failure modes:

**Uneven Anomaly Distributions**   Although ViP$^2$-CLIP consistently improves the average performance across benchmarks, its image-level performance remains less competitive on several datasets and categories. We attribute this limitation mainly to the fixed Top-$K$ aggregation strategy used in UTPA. UTPA computes global anomaly scores by aggregating the Top-$K$ anomalous patches. While this design effectively promotes unified optimization between detection and localization, the optimal aggregation scale may vary across datasets with different anomaly distributions.

To further validate this observation, we additionally evaluate different Top-$K$ values on several representative failure categories from MVTec-AD and VisA. As shown in Table 14, categories containing subtle or small-scale anomalies, such as *Cable*, *Transistor*, and *PCB3*, generally achieve better image-level AUROC under smaller aggregation scales, since smaller $K$ values better preserve localized anomaly responses during global score computation. In contrast, categories with relatively larger anomalous regions, such as *PCB1*, remain comparatively stable across different $K$ values and can benefit from larger aggregation ranges that capture broader abnormal patterns. These above analysis suggests that adaptive aggregation strategies may further improve flexibility by adjusting the aggregation scale to category-specific anomaly patterns.

**Contextual Anomalies**   The model struggles to detect contextual anomalies, such as abnormal spatial arrangements or misaligned components, which lack explicit visual distortions. As illustrated in Figure 16, these cases are difficult to resolve without prior knowledge of the normal structural layout. This limitation highlights a promising

Table 14: Sensitivity of image-level AUROC under different Top-$K$ values.

| Category | Top-10 | Top-50 | Top-100 |
|---|---|---|---|
| Cable | 76.1 | 75.3 | 73.8 |
| Transistor | 81.0 | 79.9 | 78.6 |
| PCB1 | 89.9 | 90.8 | 91.1 |
| PCB3 | 80.9 | 78.3 | 77.5 |

future direction: integrating object-level positional priors or spatial layout descriptions into the prompt design to enhance contextual reasoning.

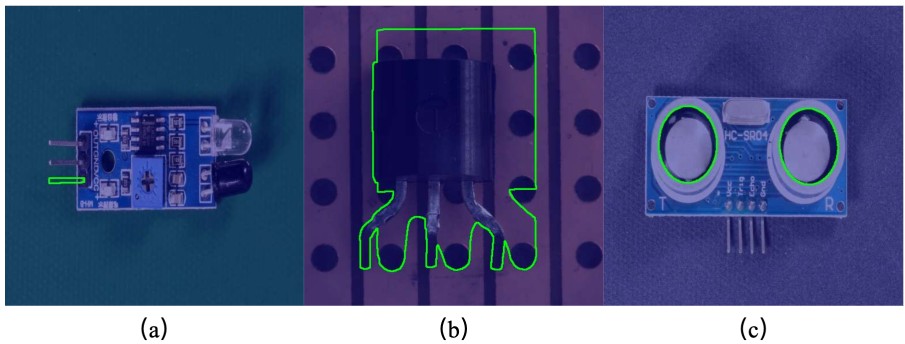

(a)            (b)            (c)

Figure 16: **Failure cases of ViP$^2$-CLIP on contextual anomalies.** Three representative categories are illustrated, where anomalies arise from irregular spatial arrangements or uneven distributions rather than explicit appearance distortions.

**Highly Textured Backgrounds**   Our method assumes the target object is centrally located against a relatively clean background, which allows the image encoder to extract a reliable global visual embedding. However, in cluttered scenes or when multiple objects are present, the quality of this extracted global embedding deteriorates. Without class-name guidance, this corrupted global context impairs prompt generation, thereby reducing localization accuracy.

Overall, these observations suggest that current prompt-based ZSAD frameworks still face challenges in modeling structural dependencies, suppressing background interference, and adapting to highly diverse anomaly distributions. Future work may benefit from integrating spatial reasoning, layout-aware prompting, and adaptive multi-scale aggregation mechanisms into unified text-patch alignment frameworks, thereby improving robustness under complex real-world anomaly scenarios.

## D   Additional Quantitative and Qualitative Results

### D.1   Fine-grained Subset Performance

Tables 15–26 report the detailed ZSAD performance at the subset level for the MVTec AD and VisA datasets. These results demonstrate ViP$^2$-CLIP's robust capability to handle fine-grained intra-class variations.

### D.2   Visualization

We provide extensive visualizations of pixel-wise anomaly maps in Figures 17–30, further demonstrating ViP$^2$-CLIP's strong segmentation capability and cross-domain generalization. The industrial examples include capsule, grid, leather, screw, and hazelnut from MVTec AD; candle, cashew, macaroni, pcb, and pipe fryum from VisA; brackets and metal plates from MPDD; as well as unknown products from BTAD. The medical examples cover melanoma detection from ISIC, and colorectal polyp detection from both CVC-ColonDB and CVC-ClinicDB.

Table 15: Fine-grained performance comparison for Pixel-level AUROC on VisA.

| Object name | CLIP | WinCLIP | APRIL-GAN | AdaCLIP | AnomalyCLIP | ViP$^2$-CLIP |
|---|---|---|---|---|---|---|
| Candle | 32.7 | 87.0 | 97.8 | 98.9 | 98.8 | 99.0 |
| Capsules | 44.8 | 80.0 | 97.5 | 98.7 | 95.0 | 98.3 |
| Cashew | 21.8 | 84.8 | 85.8 | 92.1 | 93.8 | 91.4 |
| Chewinggum | 37.8 | 95.4 | 99.5 | 99.6 | 99.3 | 99.7 |
| Fryum | 26.0 | 87.7 | 91.9 | 94.5 | 94.6 | 92.6 |
| Macaroni1 | 52.9 | 50.3 | 98.8 | 99.2 | 98.3 | 99.5 |
| Macaroni2 | 70.4 | 44.7 | 97.8 | 98.4 | 97.6 | 98.7 |
| Pcb1 | 61.9 | 38.6 | 92.8 | 93.7 | 94.0 | 90.1 |
| Pcb2 | 27.4 | 58.7 | 89.8 | 90.8 | 92.4 | 92.0 |
| Pcb3 | 71.7 | 76.0 | 88.2 | 88.5 | 88.4 | 92.2 |
| Pcb4 | 49.2 | 91.4 | 94.5 | 96.1 | 95.7 | 96.4 |
| Pipe_fryum | 26.9 | 83.6 | 96.0 | 96.0 | 98.2 | 95.2 |
| Mean | 43.6 | 73.2 | 94.2 | 95.5 | 95.5 | 95.4 |

Table 16: Fine-grained performance comparison for Pixel-level AUPRO on VisA.

| Object name | CLIP | WinCLIP | APRIL-GAN | AnomalyCLIP | AdaCLIP | ViP$^2$-CLIP |
|---|---|---|---|---|---|---|
| Candle | 7.9 | 77.7 | 92.3 | 96.5 | 62.2 | 96.7 |
| Capsules | 12.0 | 39.4 | 86.1 | 78.9 | 38.3 | 93.5 |
| Cashew | 0.3 | 78.4 | 91.5 | 91.9 | 57.5 | 94.9 |
| Chewinggum | 9.1 | 69.6 | 87.5 | 90.9 | 55.8 | 96.8 |
| Fryum | 6.1 | 74.4 | 89.4 | 86.9 | 52.7 | 94.8 |
| Macaroni1 | 25.9 | 24.8 | 93.0 | 89.8 | 65.6 | 96.4 |
| Macaroni2 | 32.6 | 8.0 | 82.0 | 84.0 | 62.9 | 89.5 |
| Pcb1 | 18.4 | 20.7 | 87.3 | 80.7 | 47.7 | 91.4 |
| Pcb2 | 8.2 | 20.7 | 75.4 | 78.9 | 50.2 | 80.0 |
| Pcb3 | 27.2 | 43.8 | 77.2 | 76.8 | 41.8 | 84.3 |
| Pcb4 | 11.6 | 74.5 | 86.6 | 89.4 | 63.9 | 90.7 |
| Pipe_fryum | 9.1 | 80.3 | 90.9 | 96.2 | 83.2 | 97.2 |
| Mean | 14.0 | 51.0 | 86.6 | 86.7 | 56.8 | 92.2 |

Table 17: Fine-grained performance comparison for Pixel-level F1 on VisA.

| Object name | CLIP | WinCLIP | APRIL-GAN | AnomalyCLIP | AdaCLIP | ViP$^2$-CLIP |
|---|---|---|---|---|---|---|
| Candle | 0.3 | 8.9 | 39.4 | 37.8 | 42.1 | 42.4 |
| Capsules | 1.1 | 4.2 | 49.1 | 37.8 | 54.1 | 50.7 |
| Cashew | 2.2 | 9.6 | 22.7 | 25.8 | 35.1 | 23.8 |
| Chewinggum | 1.1 | 31.6 | 78.5 | 61.0 | 78.2 | 74.4 |
| Fryum | 4.5 | 16.2 | 29.5 | 30.3 | 32.7 | 25.6 |
| Macaroni1 | 0.1 | 0.1 | 35.3 | 23.7 | 32.7 | 38.9 |
| Macaroni2 | 0.1 | 0.1 | 13.9 | 5.1 | 14.1 | 9.7 |
| Pcb1 | 2.3 | 0.9 | 12.2 | 12.7 | 13.7 | 13.4 |
| Pcb2 | 0.4 | 1.5 | 23.3 | 15.8 | 32.5 | 23.4 |
| Pcb3 | 1.2 | 2.1 | 21.9 | 9.3 | 35.8 | 25.7 |
| Pcb4 | 1.5 | 24.6 | 31.0 | 34.7 | 43.0 | 39.8 |
| Pipe_fryum | 2.5 | 8.3 | 30.4 | 45.5 | 29.5 | 35.9 |
| Mean | 1.5 | 9.0 | 32.3 | 28.3 | 37.0 | 33.6 |

Table 18: Fine-grained performance comparison for Image-level AUROC on VisA.

| Object name | CLIP | WinCLIP | APRIL-GAN | AnomalyCLIP | AdaCLIP | ViP$^2$-CLIP |
|---|---|---|---|---|---|---|
| Candle | 55.2 | 94.8 | 82.6 | 80.9 | 96.0 | 90.4 |
| Capsules | 61.9 | 79.4 | 62.3 | 82.8 | 87.0 | 93.8 |
| Cashew | 68.3 | 91.2 | 86.6 | 76.0 | 88.1 | 96.4 |
| Chewinggum | 59.4 | 95.5 | 96.4 | 97.2 | 93.9 | 97.9 |
| Fryum | 48.4 | 73.6 | 93.8 | 92.7 | 91.8 | 88.5 |
| Macaroni1 | 61.5 | 79.0 | 69.3 | 86.7 | 84.2 | 86.3 |
| Macaroni2 | 52.3 | 67.1 | 65.7 | 72.2 | 67.5 | 70.8 |
| Pcb1 | 66.1 | 72.1 | 51.0 | 85.2 | 89.2 | 90.8 |
| Pcb2 | 64.4 | 47.0 | 71.4 | 62.0 | 84.3 | 78.8 |
| Pcb3 | 50.7 | 63.9 | 66.9 | 61.7 | 78.9 | 78.3 |
| Pcb4 | 64.8 | 74.2 | 94.7 | 93.9 | 95.3 | 97.5 |
| Pipe_fryum | 68.9 | 67.9 | 89.2 | 92.3 | 90.3 | 92.4 |
| Mean | 60.2 | 75.5 | 77.5 | 82.0 | 87.2 | 88.5 |

Table 19: Fine-grained performance comparison for Image-level AP on VisA.

| Object name | CLIP | WinCLIP | APRIL-GAN | AnomalyCLIP | AdaCLIP | ViP$^2$-CLIP |
|---|---|---|---|---|---|---|
| Candle | 56.1 | 95.4 | 86.0 | 82.6 | 96.6 | 92.6 |
| Capsules | 74.9 | 87.9 | 74.5 | 89.4 | 92.5 | 96.7 |
| Cashew | 80.9 | 96.0 | 93.8 | 89.3 | 94.9 | 98.4 |
| Chewinggum | 77.3 | 98.2 | 98.4 | 98.8 | 97.5 | 99.1 |
| Fryum | 70.4 | 86.9 | 97.1 | 96.6 | 96.3 | 94.5 |
| Macaroni1 | 60.3 | 80.0 | 67.4 | 85.5 | 83.1 | 87.9 |
| Macaroni2 | 49.9 | 65.1 | 64.8 | 70.8 | 67.7 | 71.1 |
| Pcb1 | 69.9 | 73.0 | 55.3 | 86.7 | 89.4 | 91.0 |
| Pcb2 | 62.5 | 46.1 | 73.4 | 64.4 | 85.6 | 78.9 |
| Pcb3 | 50.6 | 63.1 | 70.4 | 69.4 | 82.4 | 81.6 |
| Pcb4 | 59.6 | 70.1 | 94.8 | 94.3 | 95.5 | 96.8 |
| Pipe_fryum | 82.1 | 82.1 | 94.5 | 96.3 | 95.1 | 96.3 |
| Mean | 66.2 | 78.7 | 80.9 | 85.3 | 89.7 | 90.4 |

Table 20: Fine-grained performance comparison for Image-level F1 on VisA.

| Object name | CLIP | WinCLIP | APRIL-GAN | AnomalyCLIP | AdaCLIP | ViP$^2$-CLIP |
|---|---|---|---|---|---|---|
| Candle | 67.4 | 90.6 | 77.9 | 75.6 | 89.5 | 83.0 |
| Capsules | 76.9 | 80.5 | 78.0 | 82.2 | 85.0 | 90.8 |
| Cashew | 80.2 | 88.9 | 85.4 | 80.3 | 86.8 | 94.0 |
| Chewinggum | 80.0 | 93.8 | 93.2 | 94.8 | 91.5 | 96.4 |
| Fryum | 80.0 | 80.0 | 91.5 | 90.1 | 88.2 | 88.2 |
| Macaroni1 | 71.5 | 74.2 | 70.8 | 80.4 | 80.4 | 79.6 |
| Macaroni2 | 66.7 | 68.8 | 69.3 | 71.2 | 68.8 | 71.3 |
| Pcb1 | 68.1 | 70.2 | 66.9 | 78.8 | 81.5 | 85.0 |
| Pcb2 | 68.4 | 67.1 | 69.1 | 67.8 | 77.5 | 74.1 |
| Pcb3 | 66.4 | 67.6 | 66.7 | 66.4 | 73.6 | 73.3 |
| Pcb4 | 69.6 | 75.7 | 87.3 | 87.8 | 89.7 | 92.4 |
| Pipe_fryum | 80.8 | 80.3 | 88.1 | 89.8 | 89.0 | 89.5 |
| Mean | 73.0 | 78.2 | 78.7 | 80.4 | 83.5 | 84.8 |

Table 21: Fine-grained performance comparison for Pixel-level AUROC on MVTec.

| Object name | CLIP | WinCLIP | VAND | AnomalyCLIP | AdaCLIP | ViP$^2$-CLIP |
|---|---|---|---|---|---|---|
| Carpet | 18.0 | 90.9 | 98.4 | 98.8 | 98.6 | 99.2 |
| Bottle | 19.6 | 85.7 | 83.5 | 90.4 | 92.8 | 89.4 |
| Hazelnut | 27.6 | 95.7 | 96.1 | 97.2 | 98.6 | 96.7 |
| Leather | 12.7 | 95.5 | 99.1 | 98.6 | 99.3 | 99.2 |
| Cable | 44.1 | 61.3 | 72.3 | 78.9 | 76.6 | 73.2 |
| Capsule | 58.0 | 87.0 | 92.0 | 95.8 | 94.3 | 94.4 |
| Grid | 11.8 | 79.4 | 95.8 | 97.3 | 91.1 | 97.7 |
| Pill | 45.6 | 72.7 | 76.2 | 91.8 | 86.7 | 87.5 |
| Transistor | 42.2 | 83.7 | 62.4 | 70.8 | 63.5 | 64.6 |
| Metal Nut | 33.3 | 49.3 | 65.5 | 74.6 | 68.2 | 76.8 |
| Screw | 72.3 | 91.1 | 97.8 | 97.5 | 97.8 | 98.7 |
| Toothbrush | 26.0 | 86.2 | 95.8 | 91.9 | 97.2 | 93.4 |
| Zipper | 52.5 | 91.7 | 91.1 | 91.3 | 95.7 | 97.0 |
| Tile | 34.6 | 79.1 | 92.7 | 94.7 | 89.5 | 93.7 |
| Wood | 36.1 | 85.1 | 95.8 | 96.4 | 94.0 | 95.7 |
| Mean | 35.6 | 82.3 | 87.6 | 91.1 | 89.6 | 90.5 |

Table 22: Fine-grained performance comparison for Pixel-level AUPRO on MVTec.

| Object name | CLIP | WinCLIP | VAND | AnomalyCLIP | AdaCLIP | ViP$^2$-CLIP |
|---|---|---|---|---|---|---|
| Carpet | 6.2 | 66.3 | 48.5 | 90.0 | 38.1 | 97.9 |
| Bottle | 0.3 | 69.9 | 45.6 | 80.8 | 39.0 | 83.5 |
| Hazelnut | 4.7 | 81.4 | 70.3 | 92.5 | 19.4 | 93.1 |
| Leather | 1.4 | 86.0 | 72.4 | 92.2 | 57.0 | 98.8 |
| Cable | 8.8 | 39.4 | 25.7 | 64.0 | 43.1 | 66.5 |
| Capsule | 31.6 | 63.7 | 51.3 | 87.6 | 60.3 | 92.7 |
| Grid | 0.2 | 49.3 | 31.6 | 75.4 | 57.9 | 89.5 |
| Pill | 5.3 | 66.9 | 65.4 | 88.1 | 40.5 | 93.1 |
| Transistor | 8.6 | 45.5 | 21.3 | 58.2 | 27.1 | 55.1 |
| Metal Nut | 0.9 | 39.6 | 38.4 | 71.1 | 63.9 | 79.8 |
| Screw | 48.5 | 70.2 | 67.1 | 88.0 | 16.1 | 94.2 |
| Toothbrush | 2.3 | 67.9 | 54.5 | 88.5 | 58.9 | 88.8 |
| Zipper | 20.8 | 72.1 | 10.7 | 65.4 | 18.1 | 89.1 |
| Tile | 6.3 | 54.5 | 26.7 | 87.4 | 25.7 | 89.2 |
| Wood | 12.9 | 56.3 | 31.1 | 91.5 | 2.4 | 95.5 |
| Mean | 10.6 | 61.9 | 44.0 | 81.4 | 37.8 | 87.1 |

Table 23: Fine-grained performance comparison for Pixel-level F1 on MVTec.

| Object name | CLIP | WinCLIP | VAND | AnomalyCLIP | AdaCLIP | ViP$^2$-CLIP |
|---|---|---|---|---|---|---|
| Carpet | 3.2 | 33.9 | 65.7 | 57.0 | 59.5 | 65.5 |
| Bottle | 10.9 | 49.4 | 53.4 | 51.6 | 32.2 | 52.3 |
| Hazelnut | 4.2 | 39.1 | 50.5 | 47.6 | 34.2 | 49.9 |
| Leather | 1.3 | 30.8 | 50.0 | 33.2 | 66.6 | 43.0 |
| Cable | 5.6 | 12.2 | 23.9 | 18.9 | 37.0 | 24.1 |
| Capsule | 4.8 | 14.3 | 33.1 | 31.0 | 63.3 | 34.4 |
| Grid | 1.4 | 13.7 | 40.8 | 32.0 | 49.8 | 41.3 |
| Pill | 7.4 | 11.8 | 27.7 | 35.5 | 31.2 | 26.0 |
| Transistor | 9.2 | 27.0 | 19.0 | 18.8 | 31.7 | 16.0 |
| Metal Nut | 21.0 | 23.8 | 28.0 | 33.1 | 28.6 | 35.1 |
| Screw | 6.4 | 11.3 | 41.7 | 33.4 | 67.4 | 46.4 |
| Toothbrush | 3.0 | 10.5 | 48.1 | 29.0 | 47.8 | 31.9 |
| Zipper | 5.2 | 27.8 | 40.5 | 45.0 | 18.0 | 53.2 |
| Tile | 13.2 | 30.8 | 66.5 | 64.9 | 52.9 | 68.0 |
| Wood | 7.4 | 35.4 | 60.3 | 55.2 | 55.6 | 58.7 |
| Mean | 6.9 | 24.8 | 43.3 | 39.1 | 45.1 | 43.1 |

Table 24: Fine-grained performance comparison for Image-level AUROC on MVTec.

| Object name | CLIP | WinCLIP | VAND | AnomalyCLIP | AdaCLIP | ViP$^2$-CLIP |
|---|---|---|---|---|---|---|
| Carpet | 86.9 | 99.3 | 99.4 | 100 | 99.9 | 99.9 |
| Bottle | 20.5 | 98.6 | 91.9 | 88.7 | 97.9 | 93.7 |
| Hazelnut | 53 | 92.3 | 89.4 | 97.2 | 97.2 | 93.5 |
| Leather | 98.5 | 100 | 99.7 | 99.8 | 99.9 | 100 |
| Cable | 51.4 | 85 | 88.1 | 70.3 | 65.5 | 75.3 |
| Capsule | 60.7 | 68.6 | 79.9 | 89.5 | 86.5 | 90.2 |
| Grid | 77 | 99.2 | 86.5 | 97.8 | 99.4 | 99.7 |
| Pill | 61.2 | 81.5 | 80.9 | 81.1 | 84.6 | 84.6 |
| Transistor | 45.6 | 89.1 | 80.8 | 93.9 | 82.2 | 79.9 |
| Metal Nut | 68.2 | 96.2 | 68.4 | 92.4 | 75.9 | 77.3 |
| Screw | 62.2 | 71.6 | 85.1 | 82.1 | 97.8 | 90 |
| Toothbrush | 23.6 | 85.3 | 53.6 | 85.3 | 93.3 | 94.2 |
| Zipper | 90 | 91.2 | 89.6 | 98.4 | 96.4 | 92.9 |
| Tile | 98.9 | 99.9 | 99.9 | 100 | 99.9 | 98.5 |
| Wood | 99.6 | 97.6 | 99 | 96.9 | 99.0 | 98.6 |
| Mean | 66.5 | 90.4 | 86.1 | 91.6 | 90.1 | 91.2 |

Table 25: Fine-grained performance comparison for Image-level AP on MVTec.

| Object name | CLIP | WinCLIP | VAND | AnomalyCLIP | AdaCLIP | ViP$^2$-CLIP |
|---|---|---|---|---|---|---|
| Carpet | 95.6 | 97.8 | 99.8 | 100 | 99.9 | 100 |
| Bottle | 65 | 97.6 | 97.6 | 96.8 | 99.4 | 98.1 |
| Hazelnut | 67.2 | 88.6 | 94.6 | 98.5 | 98.4 | 96.8 |
| Leather | 99.5 | 100 | 99.9 | 99.9 | 99.9 | 100 |
| Cable | 66.2 | 89.8 | 92.9 | 81.7 | 81.8 | 84.5 |
| Capsule | 88.5 | 90.5 | 95.4 | 97.8 | 97.1 | 97.9 |
| Grid | 90.6 | 98.2 | 94.9 | 99.3 | 99.8 | 99.9 |
| Pill | 89.9 | 96.4 | 96.1 | 95.3 | 96.6 | 96.6 |
| Transistor | 46.3 | 84.9 | 77.5 | 92.1 | 82.7 | 80 |
| Metal Nut | 91.1 | 99.1 | 91.8 | 98.2 | 94.5 | 94.7 |
| Screw | 81.6 | 87.7 | 93.6 | 92.9 | 96.5 | 96.5 |
| Toothbrush | 60.7 | 94.5 | 71.6 | 93.9 | 97.7 | 97.5 |
| Zipper | 96.6 | 97.5 | 97.1 | 99.5 | 99.1 | 98.2 |
| Tile | 99.5 | 100 | 100 | 100 | 100 | 99.5 |
| Wood | 99.9 | 99.3 | 99.7 | 99.2 | 99.7 | 99.6 |
| Mean | 82.6 | 95.6 | 93.5 | 96.4 | 95.6 | 96 |

Table 26: Fine-grained performance comparison for Image-level F1 on MVTec.

| Object name | CLIP | WinCLIP | VAND | AnomalyCLIP | AdaCLIP | ViP$^2$-CLIP |
|---|---|---|---|---|---|---|
| Carpet | 89.4 | 97.8 | 98.3 | 99.4 | 99.4 | 99.4 |
| Bottle | 86.3 | 97.6 | 92.1 | 90.9 | 92.4 | 92.4 |
| Hazelnut | 77.8 | 88.6 | 87 | 92.6 | 91.2 | 91.2 |
| Leather | 96.7 | 100 | 98.9 | 99.5 | 99.5 | 100 |
| Cable | 76 | 84.8 | 84.5 | 77.4 | 76.0 | 80.2 |
| Capsule | 90.5 | 93.5 | 91.5 | 91.7 | 93.1 | 93 |
| Grid | 88.4 | 98.2 | 89.1 | 97.3 | 98.3 | 99.1 |
| Pill | 91.6 | 91.6 | 91.6 | 92.1 | 91.8 | 91.8 |
| Transistor | 57.1 | 80 | 73.9 | 83.7 | 75.9 | 71.8 |
| Metal Nut | 89.4 | 95.3 | 89.4 | 93.7 | 89.4 | 90.3 |
| Screw | 85.6 | 85.9 | 89.3 | 88.3 | 91.3 | 91.3 |
| Toothbrush | 83.3 | 88.9 | 83.3 | 90 | 93.5 | 93.5 |
| Zipper | 91.5 | 93.4 | 90.8 | 97.9 | 95.1 | 92.4 |
| Tile | 97 | 99.4 | 99.4 | 100 | 99.4 | 96.4 |
| Wood | 99.2 | 95.2 | 97.4 | 96.6 | 96.7 | 96.8 |
| Mean | 86.7 | 92.7 | 90.4 | 92.7 | 92.3 | 92 |

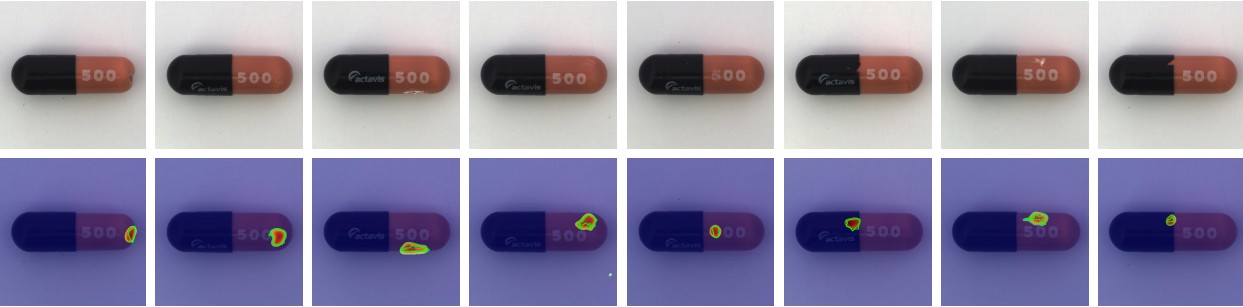

Figure 17: Anomaly score maps for the data subset, capsule, in MVTec AD. The first row represents the input, the second row presents the segmentation results from ViP$^2$-CLIP.

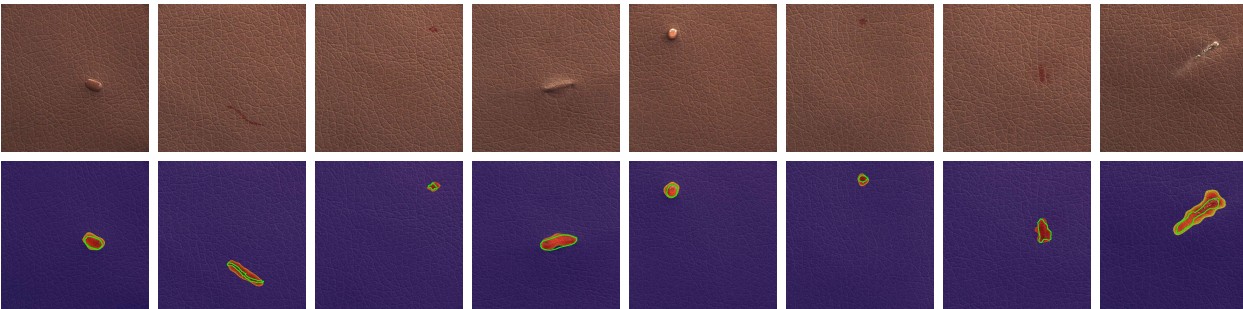

Figure 18: Anomaly score maps for the data subset, leather, in MVTec AD. The first row represents the input, the second row presents the segmentation results from ViP$^2$-CLIP.

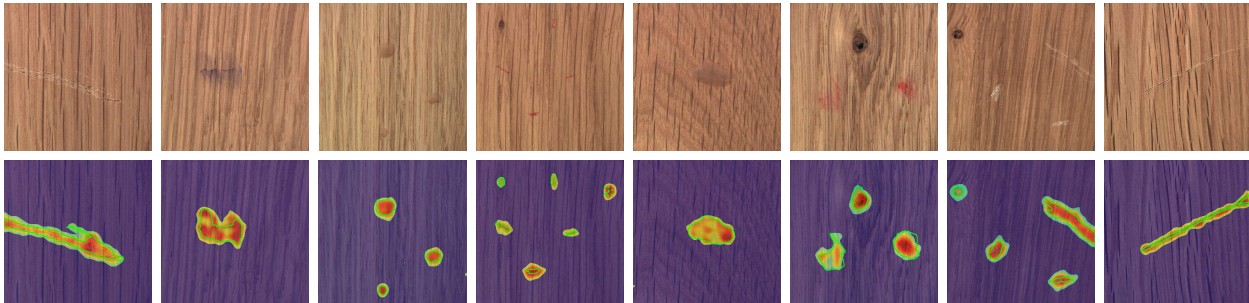

Figure 19: Anomaly score maps for the data subset, wood, in MVTec AD. The first row represents the input, the second row presents the segmentation results from ViP$^2$-CLIP.

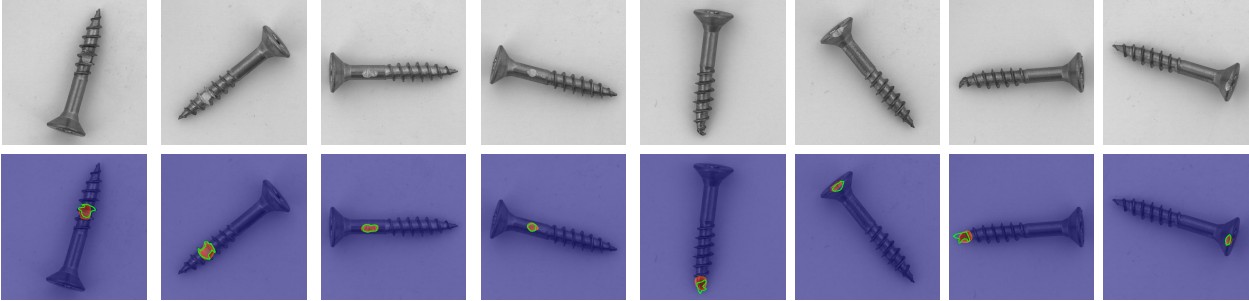

Figure 20: Anomaly score maps for the data subset, screw, in MVTec AD. The first row represents the input, the second row presents the segmentation results from ViP$^2$-CLIP.

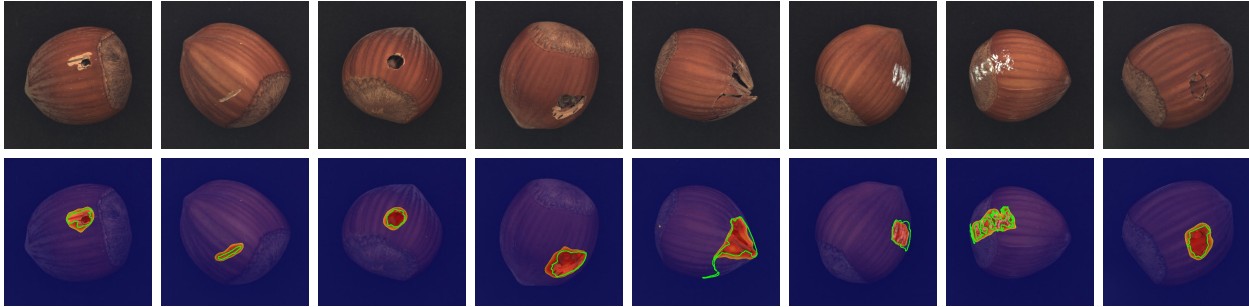

Figure 21: Anomaly score maps for the data subset, hazelnut, in MVTec AD. The first row represents the input, the second row presents the segmentation results from ViP$^2$-CLIP.

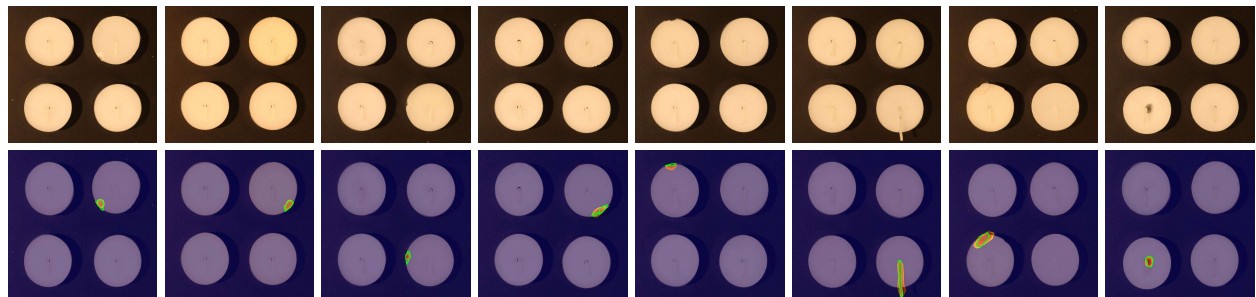

Figure 22: Anomaly score maps for the data subset, candle, in VisA. The first row represents the input, the second row presents the segmentation results from ViP$^2$-CLIP.

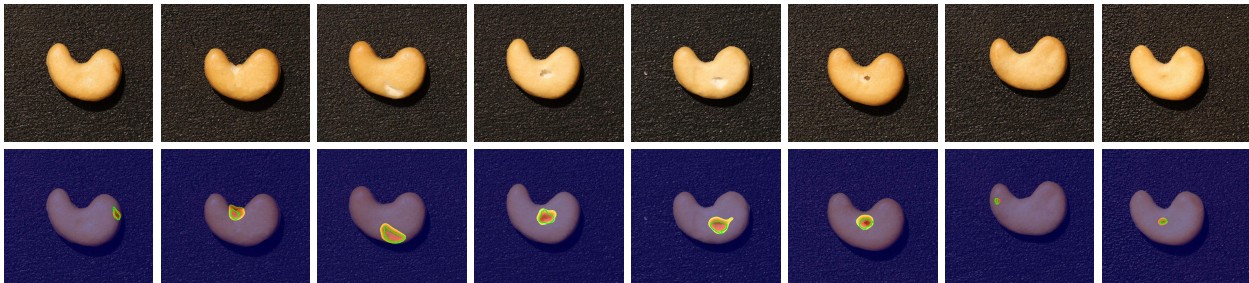

Figure 23: Anomaly score maps for the data subset, cashew, in VisA. The first row represents the input, the second row presents the segmentation results from ViP²-CLIP.

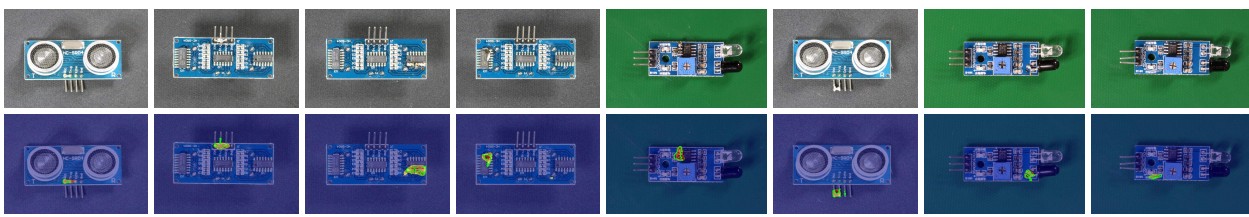

Figure 24: Anomaly score maps for the data subset, pcb, in VisA. The first row represents the input, the second row presents the segmentation results from ViP²-CLIP.

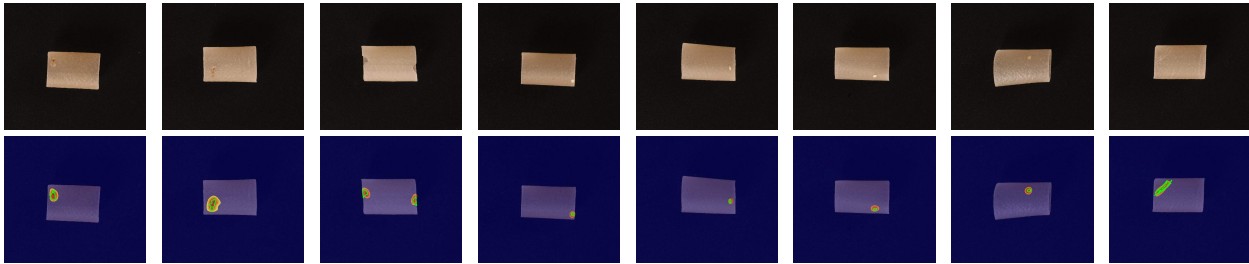

Figure 25: Anomaly score maps for the data subset, pipe fryum, in VisA. The first row represents the input, the second row presents the segmentation results from ViP²-CLIP.

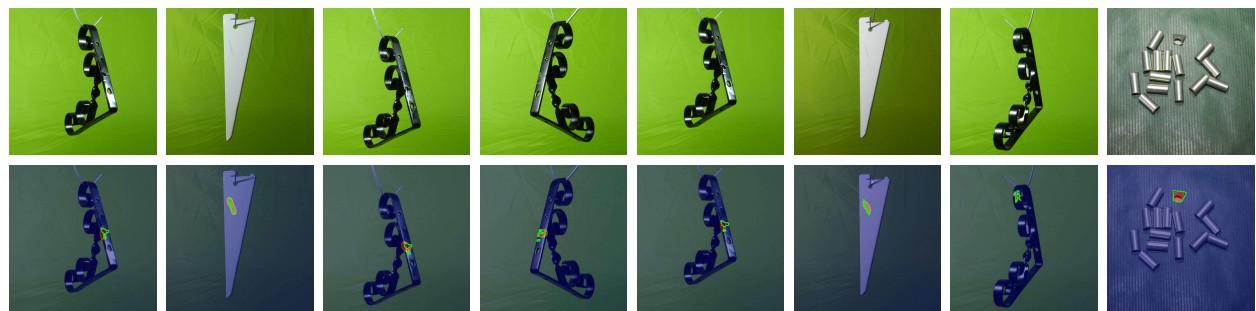

Figure 26: Anomaly score maps for the data subset in MPDD. The first row represents the input, the second row presents the segmentation results from ViP²-CLIP.

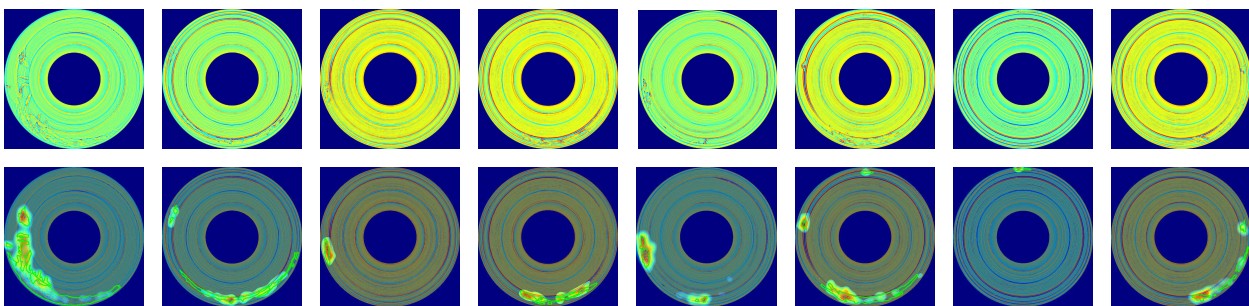

Figure 27: Anomaly score maps for the data subset in BTAD. The first row represents the input, the second row presents the segmentation results from ViP$^2$-CLIP.

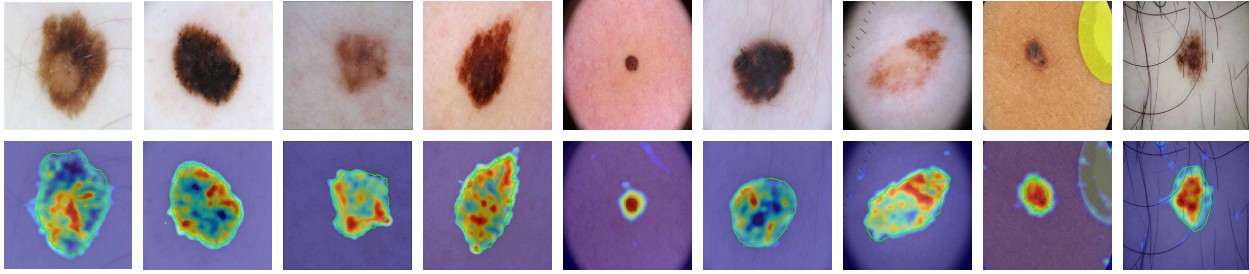

Figure 28: Anomaly score maps for the data subset skin. The first row represents the input, the second row presents the segmentation results from ViP$^2$-CLIP.

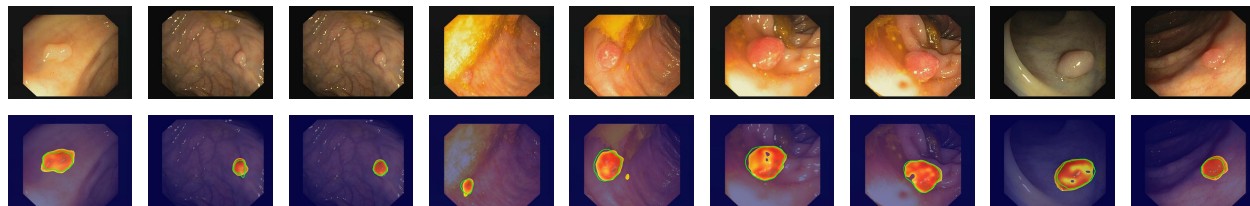

Figure 29: Anomaly score maps for the data subset colon. The first row represents the input, the second row presents the segmentation results from ViP$^2$-CLIP.

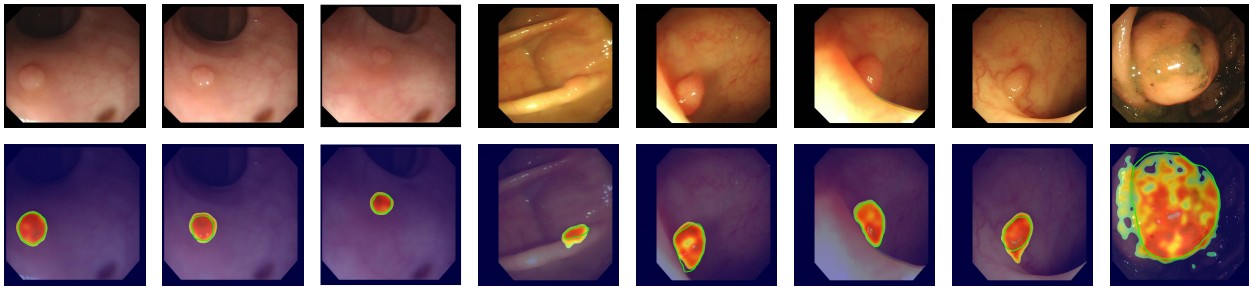

Figure 30: Anomaly score maps for the data subset colon. The first row represents the input, the second row presents the segmentation results from ViP$^2$-CLIP.

