# OpenReview forum: "ViP$^2$-CLIP: Visual-Perception Prompting with Unified Alignment for Zero-Shot Anomaly Detection"
_TMLR — Decision pending for TMLR_

### Review · Reviewer_LH7f · 2026-05-04

**Summary Of Contributions:**

This paper proposes ViP^2-CLIP, a lightweight framework for Zero-Shot Anomaly Detection (ZSAD) based on the CLIP model. The authors identify a critical limitation in existing CLIP-based ZSAD methods: their high sensitivity to prompt wording and reliance on explicit category labels, which hinders robustness in scenarios with ambiguous or privacy-constrained labels. To address this, ViP$^2$-CLIP introduces two main components, Visual-Perception Prompting (ViP-Prompt) and Unified Text-Patch Alignment (UTPA).

**Key Strengths:**
*   The method effectively eliminates the need for manual templates and class-name priors by using visual-conditioned prompts. Experiments show it remains stable even when class names are swapped or ambiguous, unlike baselines like WinCLIP and AdaCLIP.
*   The UTPA strategy successfully balances image-level detection and pixel-level localization by unifying the alignment objective, avoiding the trade-off issues seen in dual-branch architectures.
*   The framework is lightweight, adding only two adapters to a frozen CLIP backbone. It demonstrates superior computational efficiency, requiring significantly less training time (0.54h vs. 5.62h for FAPrompt on VisA) and fewer trainable parameters (4.15M) compared to state-of-the-art competitors.
*   Evaluated on 14 industrial and medical benchmarks, ViP$^2$-CLIP achieves competitive or state-of-the-art performance, particularly in pixel-level metrics like PRO and F1.

**Key Weaknesses:**
*   While generally competitive, ViP$^2$-CLIP does not always outperform baselines in every metric. For example, on MVTec AD image-level AUROC, AnomalyCLIP scores 91.6% compared to ViP$^2$-CLIP's 91.2%.
*   The authors acknowledge that the method struggles with contextual anomalies (e.g., spatial arrangement errors) because it relies on visual distortions rather than structural priors.
*   Performance on medical datasets drops when not fine-tuned on domain-specific auxiliary data, highlighting a remaining limitation in cross-domain generalization without specific fine-tuning.

**Audience:**

Yes

**Audience Explanation:**

This work is highly relevant to the TMLR audience, specifically researchers in Vision-Language Models, Anomaly Detection, Efficient Model Adaptation.

**Broader Impact Concerns:**

None.

**Claims And Evidence:**

Yes

**Claims Explanation:**

The authors provide extensive quantitative comparisons against six leading SOTA baselines across 14 benchmarks, along with detailed ablation studies that isolate the contributions of ViP-VCA, ViP-FGP, and UTPA. Sensitivity analyses for key hyperparameters (e.g., Top-K selection, prompt length, and state adjectives) further substantiate the design choices. Qualitative visualizations and honest discussions of failure cases (e.g., contextual anomalies and complex backgrounds) enhance the credibility and transparency of the results

**Requested Changes:**

*   The authors should analyze why ViP$^2$-CLIP underperforms on certain specific metrics or datasets compared to strong baselines (e.g., image-level AUROC on MVTec AD or VisA). Currently, the paper highlights overall average improvements but lacks a discussion on the trade-offs or limitations that lead to these specific gaps.
*   While the Top-$K$ ablation is provided, a brief discussion on the sensitivity of the number of prompt tokens or the specific layers chosen for UTPA alignment would enhance reproducibility.
*   The authors should briefly discuss potential future directions, such as integrating spatial layout priors, to mitigate the identified failure cases with contextual anomalies.
*   Provide a deeper analysis of the performance gap between industrial and medical datasets, potentially including feature distribution comparisons or attention weight visualizations to clarify the impact of domain shifts.

---

> ### Author Response · Authors · 2026-05-21
>
> We sincerely thank you for your insightful comments, which have helped improve the clarity and quality of our manuscript. Below, we provide point-by-point responses and summarize the corresponding revisions made in the updated version. We would be happy to provide any additional information if needed and sincerely look forward to your further feedback.
>
> **Note: In the revised version, all modifications are highlighted in blue.**
>
> > comments 1: The authors should analyze why ViP-CLIP underperforms on certain specific metrics or datasets compared to strong baselines. Currently, the paper highlights overall average improvements but lacks a discussion on the trade-offs or limitations that lead to these specific gaps.
>
> Thank you for this valuable comments. Following your suggestion, we conducted a more detailed analysis on representative failure cases from MVTec AD and VisA datasets.
>
> Based on the per-category results in Appendix Tables 14–25, we observe that the main performance gaps mainly occur in categories containing extremely small or subtle defects, such as Cable and Transistor in MVTec AD, and PCBs in VisA (where image-level metrics are relatively lower). We attribute this behavior mainly to the fixed Top-$K$ aggregation strategy used in UTPA. Specifically, UTPA derives image-level anomaly scores by aggregating the most suspicious local patch responses, which effectively promotes consistent optimization between localization and detection. However, for categories with tiny or low-contrast defects, a fixed Top-$K$ selection may include additional normal patches during global aggregation, thereby weakening the anomaly signal for image-level scoring.
>
> Nevertheless, this unified aggregation strategy still provides strong and consistent pixel-level localization capability by encouraging the model to focus on highly discriminative anomalous regions. We further note that the adaptive aggregation strategies may further improve flexibility by adjusting the aggregation scale to category-specific anomaly patterns, which we consider an important direction for future work.
>
> Following your suggestion, we have explicitly discussed this trade-off in Sec. 5.2 and provide a detailed comparative analysis in Appendix C.5.
>
> > comments 2: A brief discussion on the sensitivity of the number of prompt tokens or the specific layers chosen for UTPA alignment would enhance reproducibility.
>
> Thank you for this valuable suggestion. We apologize for not making these implementation details sufficiently clear in the original manuscript.
>
> Regarding the prompt token configuration, we already provide a detailed prompt-length analysis in Appendix C.2. Specifically, we evaluate different combinations of static and dynamic prompt tokens on the VisA dataset. The results show that increasing the number of prompt tokens does not consistently improve performance. Excessively long prompts tend to introduce redundant semantics and may reduce generalization ability. Based on this analysis, we adopt a default prompt length of 10, consisting of 3 dynamic tokens and 7 static learnable tokens, which provides a good balance between semantic expressiveness and robustness.
>
> Regarding the choice of UTPA alignment layers, we follow the commonly adopted multi-scale feature extraction strategy (by selecting layers 6, 12, 18, and 24 from the CLIP visual encoder[1][2][3]). These layers are uniformly distributed across different network depths, enabling the framework to capture both fine-grained texture anomalies and higher-level semantic structures. The effectiveness of this design is further supported by the layer-wise visualization analysis in Fig. 7, where shallow and intermediate layers primarily respond to local texture anomalies, while deeper layers provide stronger semantic localization for structurally complex objects. Aggregating features across multiple depths therefore leads to more robust anomaly representations across diverse defect patterns.
>
> To improve reproducibility and clarity, we have further clarified these design choices and analyses in Appendix C.2 and the corresponding discussion of Fig. 7 in the revised manuscript.

---

> ### Author Response · Authors · 2026-05-21
>
> > comments 3: The authors should briefly discuss potential future directions, such as integrating spatial layout priors, to mitigate the identified failure cases with contextual anomalies.
>
> Thank you for your suggestion. Following your recommendation, we have added a discussion on potential future directions in the last paragraph of Appendix C.5.
>
> Specifically, we now discuss that current prompt-based ZSAD frameworks still have limited capability in handling structural anomalies, cluttered environments, and highly variable anomaly scales. To address these limitations, we further discuss that incorporating spatial layout priors and adaptive aggregation strategies into the unified text-patch alignment framework may improve robustness in complex anomaly scenarios. In particular, adaptive aggregation strategies may further enhance flexibility by adjusting the aggregation scale to category-specific anomaly patterns, which we consider an important direction for future work.
>
> > comments 4: Provide a deeper analysis of the performance gap between industrial and medical datasets, potentially including feature distribution comparisons or attention weight visualizations to clarify the impact of domain shifts.
>
> Thank you for this valuable suggestion. Following your recommendation, we added a more detailed analysis of the domain gap between industrial and medical datasets in Appendix C.4, including feature representation analysis and attention visualization results (Fig. 15).
>
> Specifically, we analyze how the substantial differences in anomaly characteristics between industrial and medical domains affect prompt-guided text-patch alignment. Industrial anomalies are typically characterized by clear structural defects and repetitive texture corruptions, whereas medical anomalies often exhibit low contrast, ambiguous boundaries, and large intra-class variations. These domain discrepancies cause substantial differences in the global visual features extracted from CLIP, thereby reducing the generalizability of visual-conditioned adapters learned from industrial dataset.
>
> To further illustrate this effect, we visualize the attention maps of normal and anomalous prompts on CVC-ClinicDB under two fine-tuning settings: (1) training with industrial auxiliary data only (MVTec AD), and (2) additional fine-tuning with medical auxiliary data (CVC-ColonDB). The results show that training only with industrial auxiliary data produces spatially diffuse and less discriminative attention maps on medical images. In contrast, introducing medical auxiliary data significantly improves attention concentration around lesion regions, leading to more accurate cross-domain localization and detection.
>
> These additional analyses further demonstrate that domain-relevant auxiliary data plays an important role in improving the generalization capability of cross-domain anomaly detection. We sincerely appreciate your insightful suggestion, which helped improve the completeness and clarity of our domain shift analysis.
>
> **References**
>
> [[1]](https://openaccess.thecvf.com/content/CVPR2023/html/Jeong_WinCLIP_Zero-Few-Shot_Anomaly_Classification_and_Segmentation_CVPR_2023_paper.html) WinCLIP: Zero-/Few-Shot Anomaly Classification and Segmentation
>
> [[2]](https://openreview.net/forum?id=buC4E91xZE) AnomalyCLIP: Object-agnostic Prompt Learning for Zero-shot Anomaly Detection
>
> [[3]](https://openaccess.thecvf.com/content/ICCV2025/html/Zhu_Fine-grained_Abnormality_Prompt_Learning_for_Zero-Shot_Anomaly_Detection_ICCV_2025_paper.html) Fine-grained Abnormality Prompt Learning for Zero-Shot Anomaly Detection

---

### Review · Reviewer_Fphh · 2026-05-11

**Summary Of Contributions:**

This paper proposes ViP2-CLIP, a CLIP-based framework for zero-shot anomaly detection that introduces two main technical contributions. The first is Visual-Perception Prompting (ViP-Prompt), which replaces fixed class-name tokens in text prompts with image-conditioned cues generated through two sub-modules: a Visual-Conditioned Adapter (ViP-VCA) that projects global image features into the prompt embedding space, and a Fine-Grained Perception module (ViP-FGP) that uses cross-attention between text prompts and multi-scale patch features to inject local visual details. The second contribution is Unified Text-Patch Alignment (UTPA), which derives both image-level anomaly scores and pixel-level anomaly maps from local patch features alone — using Top-K pooling over the most anomalous patches for the image-level score. Experiments are conducted across 14 benchmarks in industrial and medical domains, comparing against six baselines.

**Audience:**

Yes

**Audience Explanation:**

Yes. Zero-shot anomaly detection is a practically important problem with applications in manufacturing quality control and medical imaging, both areas where labeled anomaly data is scarce or expensive. The paper addresses a real and well-motivated limitation of existing CLIP-based ZSAD methods sensitivity to class-name wording, which is relevant to practitioners deploying these systems. The audience interested in vision-language models, prompt learning, and industrial/medical anomaly detection would find value in the proposed solutions. The breadth of evaluation (14 benchmarks across two domains) also increases the paper's relevance.

**Broader Impact Concerns:**

No significant broader impact concerns. The application domains (industrial defect detection and medical anomaly screening) are socially beneficial.

**Claims And Evidence:**

No

**Claims Explanation:**

1.The claimed "superiority" is overstated relative to actual margins. The paper claims to achieve "superior performance over existing state-of-the-art approaches," but the improvements are often marginal and inconsistent. On the industrial image-level average (Table 1), ViP2-CLIP edges out FAPrompt by ~0.7% AUROC and ~1.2% AP, but actually underperforms AdaCLIP on DTD-Synthetic image-level metrics and ties or loses on several individual datasets. On medical image-level (Table 2), ViP2-CLIP's average AUROC (95.1%) is lower than FAPrompt's (95.7%) and comparable to AdaCLIP's (94.7%). The pixel-level improvements are more convincing (especially PRO gains), but the narrative should be more nuanced.

2.Several relevant concurrent methods discussed in the related work section are absent from the experimental comparison. The paper's language claims "superiority" where "competitiveness with specific strengths in pixel-level PRO" would be more accurate.

3.The method also has acknowledged limitations around contextual anomalies and cluttered backgrounds that are not deeply analyzed.

**Requested Changes:**

1.Tone down claims of "superiority" throughout the paper. The results show competitive performance with particular strengths in pixel-level PRO, not consistent dominance. Phrases like "achieves superior performance over existing state-of-the-art" should be replaced with more precise characterizations.

2.Expand the motivation analysis in Appendix A to cover all (or a random majority of) MVTec AD categories, not just 3 cherry-picked ones per method. Report aggregate statistics (mean and standard deviation of metric change across synonym swaps).

3.Compare concurrent methods discussed in the related work and more deeply analyzed limitations should be added.

---

> ### Author Response · Authors · 2026-05-21
>
> We sincerely thank you for your insightful comments, which have helped improve the clarity and quality of our manuscript. Below, we provide point-by-point responses and summarize the corresponding revisions made in the updated version. We would be happy to provide any additional information if needed and sincerely look forward to your further feedback.
>
> **Note: In the revised version, all modifications are highlighted in blue.**
>
> > comments 1: Tone down claims of "superiority" throughout the paper.
>
> Thank you for this valuable suggestion. We have carefully revised the manuscript to adopt more precise and balanced descriptions of our results. We replaced overly strong claims such as “superior performance” with more accurate expressions including “competitive performance” and “strengths in pixel-level localization.” These revisions have been applied throughout the manuscript, including the Abstract, Sec. 5.2, and the Conclusion section, to ensure that the claims are fully aligned with the empirical evidence.
>
> In addition, we have refined several overly strong expressions (e.g., “resolves” and “addresses”) with more conservative wording to better align with the experimental evidence. Related revisions were incorporated into the Introduction, Related Work, and Section 5.3.
>
> > comments 2: Expand the motivation analysis in Appendix A to cover all MVTec AD categories. Report aggregate statistics.
>
> Following your suggestion, We have extended the synonym-substitution experiments to all  categories of the MVTec AD dataset and additionally report the mean and standard deviation of the performance changes across different synonym replacements, as summarized in Table 1.
>
> **Table 1: Sensitivity analysis under synonym substitutions across all 15 MVTec AD categories**
>
> Results marked with `*` are obtained using the training-free WinCLIP model, while the remaining categories are evaluated using the finetuned AdaCLIP model.
>
> | Category       | Pixel-level AUROC | Pixel-level AUPRO | Pixel-level F1 | Image-level AUROC | Image-level AP | Image-level F1 |
> |----------------|-------------------|--------------------|----------------|-------------------|----------------|----------------|
> | bottle*        | 85.2 ± 0.7        | 67.8 ± 1.9         | 48.3 ± 1.2     | 98.6 ± 0.2        | 99.5 ± 0.1     | 97.1 ± 0.4     |
> | wood*          | 85.8 ± 1.2        | 59.5 ± 4.3         | 36.3 ± 1.3     | 98.3 ± 0.8        | 99.5 ± 0.2     | 96.2 ± 1.1     |
> | zipper*        | 86.0 ± 6.0        | 63.2 ± 9.2         | 21.8 ± 6.2     | 92.2 ± 2.2        | 97.8 ± 0.6     | 94.0 ± 1.3     |
> | cable*         | 61.5 ± 0.3        | 38.8 ± 0.6         | 12.6 ± 0.8     | 84.4 ± 0.7        | 89.1 ± 0.7     | 83.9 ± 1.0     |
> | capsule*       | 86.2 ± 3.2        | 65.2 ± 1.9         | 15.6 ± 1.4     | 83.7 ± 1.1        | 90.6 ± 0.4     | 85.7 ± 0.6     |
> | transistor*    | 77.1 ± 7.3        | 46.6 ± 1.3         | 24.5 ± 2.2     | 84.3 ± 1.9        | 87.9 ± 0.8     | 82.7 ± 0.9     |
> | hazelnut*      | 93.4 ± 5.1        | 54.8 ± 8.5         | 55.6 ± 4.6     | 96.7 ± 0.7        | 97.8 ± 0.6     | 93.6 ± 0.9     |
> | metal_nut*     | 63.3 ± 4.9        | 27.2 ± 4.2         | 36.9 ± 3.7     | 74.8 ± 1.1        | 93.7 ± 1.0     | 88.4 ± 1.2     |
> | screw*         | 92.8 ± 5.0        | 24.3 ± 4.1         | 59.6 ± 4.0     | 68.9 ± 1.0        | 85.4 ± 1.1     | 86.1 ± 1.1     |
> | toothbrush     | 82.4 ± 21.8       | 45.5 ± 3.7         | 56.7 ± 2.6     | 92.4 ± 1.4        | 97.2 ± 0.6     | 90.8 ± 1.0     |
> | grid           | 88.5 ± 5.7        | 33.8 ± 5.6         | 43.4 ± 4.1     | 99.3 ± 0.5        | 99.6 ± 0.4     | 98.2 ± 0.9     |
> | leather        | 97.1 ± 3.7        | 47.1 ± 4.1         | 56.5 ± 3.2     | 99.7 ± 0.3        | 99.8 ± 0.3     | 99.2 ± 0.4     |
> | tile           | 89.0 ± 0.7        | 67.6 ± 0.1         | 16.9 ± 1.3     | 100.0 ± 0.0       | 100.0 ± 0.0    | 99.8 ± 0.3     |
> | carpet         | 97.9 ± 1.0        | 66.6 ± 0.5         | 40.9 ± 26.1    | 100.0 ± 0.0       | 100.0 ± 0.0    | 99.6 ± 0.3     |
> | pill           | 86.9 ± 0.2        | 31.7 ± 0.1         | 28.3 ± 1.8     | 89.2 ± 0.3        | 97.7 ± 0.0     | 94.4 ± 0.6     |
>
> The expanded experiments further support our conclusion. Image-level classification metrics remain relatively stable under synonym substitutions, while pixel-level localization shows significantly larger fluctuations across most categories. In several cases (e.g., carpet or toothbrush), the variation exceeds 20 percentage points, indicating that CLIP-based localization is much more sensitive to class-name perturbations. Although fine-tuned methods such as AdaCLIP improve overall performance, this sensitivity is not fully resolved.
>
> Following your suggestion, We have added the detailed per-category synonym-substitution results in Tables 8–12 of Appendix A and further included the aggregate statistics in Table 7.

---

> > ### Author Response · Authors · 2026-05-21
> >
> > > comments 3: Compare concurrent methods discussed in the related work and more deeply analyzed limitations should be added.
> >
> > Thank you for the valuable suggestion.
> >
> > **(i) Compare concurrent methods discussed in the related work.**
> >
> > For methods mentioned in the Related Work but not included in our comparisons, we clarify the following: GenCLIP is excluded due to the absence of publicly available code, while ACD-CLIP has not yet been officially released. In addition, VCP-CLIP primarily focuses on segmentation-only settings, making it less suitable for evaluating the effectiveness of the proposed UTPA framework.
> >
> > More importantly, existing CLIP-based prompt learning methods for ZSAD can generally be categorized into four representative paradigms[1]: (1) training-free prompt ensembles (e.g., WinCLIP), (2) class-name-dependent encoder fine-tuning methods (e.g., APRIL-GAN, AdaCLIP, and AA-CLIP), (3) object-agnostic prompt learning approaches (e.g., AnomalyCLIP), and (4) vision-guided learnable prompting methods (e.g., FAPrompt). Our experiments already include representative methods from each category, thereby providing a comprehensive and fair evaluation of the proposed framework. Compared with these paradigms, ViP$^2$-CLIP simultaneously removes explicit class-name priors, introduces vision-guided prompt adaptation, and unifies image-level detection and pixel-level localization through a shared text-patch alignment strategy.
> >
> > We have further clarified these distinctions in the revised Related Work section.
> >
> > **(ii) more deeply analyzed limitations.**
> >
> > Thank you for this valuable suggestion. We have added a more detailed limitation analysis in Appendix C.5. Specifically, we discuss three representative failure scenarios in the original paper: (1) uneven anomaly distributions, where anomaly scales vary significantly across categories; (2) contextual anomalies, which commonly occur in ZSAD frameworks due to the absence of explicit normal structural modeling[2]; and (3) highly textured or cluttered backgrounds, where multiple foreground objects may introduce target ambiguity and make it difficult for the model to consistently focus on the primary detection object, thereby reducing anomaly localization accuracy (which is not a primary concern in our target scenarios).
> >
> > Among these limitations, uneven anomaly distributions are most closely related to our framework. We attribute this issue to the fixed Top-$K$ setting in UTPA. Specifically, Top-$K$ pooling aggregates the $K$ most anomalous patches to compute the image-level anomaly score. A small $K$ may fail to capture spatially distributed defects, while a large $K$ may dilute anomaly signals with normal background regions.
> >
> > To further investigate this behavior, we conduct additional experiments on representative failure cases from MVTec-AD and VisA under different Top-$K$ settings. As shown in Table 2, categories with subtle or small-scale defects, such as Cable, Transistor, and PCB3, achieve better image-level AUROC under smaller Top-$K$ values. In contrast, categories containing relatively larger anomalous regions, such as PCB1, remain comparatively stable and can benefit from larger aggregation ranges. These observations support the effectiveness of the proposed unified text-patch alignment strategy, while also revealing that anomaly distributions of different spatial scales may require different aggregation behaviors.
> >
> > **Table 2. Sensitivity of image-level AUROC under different Top-K values**
> >
> > | Category   | Top-10 | Top-50 | Top-100 |
> > |------------|--------|--------|---------|
> > | Cable      | 76.1   | 75.3   | 73.8    |
> > | Transistor | 81.0   | 79.9   | 78.6    |
> > | PCB1       | 89.9   | 90.8   | 91.1    |
> > | PCB3       | 80.9   | 78.3   | 77.5    |
> >
> > In practice, we use a unified setting of $K=50$, which achieves consistently competitive performance across both MVTec-AD and VisA without dataset-specific tuning, and it generalizes well to other datasets with competitive results. However, The above analysis suggests that adaptive aggregation strategies may further improve flexibility by adjusting the aggregation scale to category-specific anomaly patterns. We consider this a promising direction for future work.
> >
> > Following your suggestion, we have added these limitation analyses and discussions in Appendix C.5 to provide a more comprehensive understanding of the proposed framework and its practical limitations.
> >
> > **References**
> >
> > [[1]](https://link.springer.com/article/10.1007/s10462-025-11287-7) A Survey of Deep Learning for Industrial Visual Anomaly Detection
> >
> > [[2]](https://link.springer.com/chapter/10.1007/978-3-031-73039-9_8) AdaCLIP: Adapting CLIP with Hybrid Learnable Prompts for Zero-Shot Anomaly Detection

---

### Review · Reviewer_A6pY · 2026-05-11

**Summary Of Contributions:**

The paper proposes ViP2-CLIP, a CLIP-based framework for Zero-Shot Anomaly Detection (ZSAD) with two components:

  1. Visual-Perception Prompting (ViP-Prompt): replaces class-name tokens with learnable tokens that are fused with (i) a
  CoCoOp-style global image conditioning obtained from a Meta-Net (ViP-VCA) and (ii) cross-attention between prompts and
  multi-scale patch features (ViP-FGP), producing layer-specific text prompts.
  2. Unified Text-Patch Alignment (UTPA): abandons the dual-branch (global-image vs. patch) alignment used by
  AnomalyCLIP/APRIL-GAN; the image-level score is obtained by Top-K pooling over patch-level similarities to the anomaly
  prompt, so detection and localization share a single alignment target.

  The method is trained on the test split of MVTec-AD (or VisA) and evaluated zero-shot on 14 industrial and medical
  benchmarks.

###  Strengths.
  - A clear, well-motivated story that ties together two previously separate observations: (i) CLIP's segmentation quality is sensitive to class-name wording (Appendix A); (ii) dual-branch image/patch alignment creates an optimization conflict (Table 5).
  - The framework is noticeably lighter than FAPrompt/AdaCLIP/AA-CLIP in trainable parameters and inference time (Table 3), which is a legitimate practical contribution.
  - Broad evaluation across 14 benchmarks, with per-category breakdowns in the appendix and a helpful failure-mode discussion (Appendix C.4).
  - The "fewer knobs" design is attractive: two light adapters on a frozen CLIP backbone.

###  Weaknesses.
  - Technical novelty is limited. ViP-VCA is essentially CoCoOp applied to the ZSAD prompt template; ViP-FGP is standard text–patch cross-attention (conceptually close to AnomalyCLIP's DPAM and to DenseCLIP); Top-K pooling over patch anomaly scores for image-level classification is a well-known MIL-style aggregator in the anomaly-detection literature. The paper does not position UTPA against these precedents.
  - The headline claim of "superior performance over existing state-of-the-art" is overstated. On several averages (e.g., MVTec-AD image-level, medical image-level) ViP2-CLIP ties or loses to WinCLIP/FAPrompt/AdaCLIP. Absolute gains on average are typically < 1% AUROC.
  - Main tables contain no error bars or seed variation, so many of the reported rankings are indistinguishable from noise. Only inference latency has a ± term.
  - The "optimization conflict" claim (Table 5) is only shown against a stripped-down CLIP_DUAL baseline built by the authors, not against the actual dual-branch architectures of AnomalyCLIP/APRIL-GAN. That weakens the causal interpretation.
  - The robustness-to-wording motivation (Appendix A) is compelling, but there is no direct quantitative comparison showing that ViP2 CLIP is more stable than class-name–based baselines under label perturbation — the link between motivation and contribution is left implicit.
  - Fixed Top-K=50 is a hard-coded assumption about anomaly spatial extent; ablation is coarse (6 points on 2 datasets) and the authors themselves list "uneven anomaly distribution" as a failure mode.
  - The "training on the test split" protocol (inherited from AnomalyCLIP) is unusual and should be discussed more carefully for a TMLR audience.

**Audience:**

Yes

**Audience Explanation:**

ZSAD is an active sub-field with steady activity at CVPR/ICCV/ECCV/ICLR. Readers working on CLIP adaptation, prompt learning, and industrial/medical anomaly detection will benefit from: (i) a clean, reproducible baseline that removes class-name dependence with two lightweight adapters; (ii) a concrete demonstration that merging the image-level and pixel-level alignment objectives into a single patch-level objective with Top-K pooling gives balanced detection/localization at lower compute; (iii) the appendix's class-name sensitivity analysis, which is useful independent of the proposed method. Even if the method does not unambiguously dominate the SOTA, the engineering and the motivation study are valuable data points for the community.

**Broader Impact Concerns:**

The paper does not include a Broader Impact Statement, while I think it necessary to state something like medical deployment risk and privacy-related issues.

**Claims And Evidence:**

No

**Claims Explanation:**

The paper's secondary claims — that the method is lightweight, label-agnostic, and avoids manual template design — are convincingly supported (Tables 3, 6; Appendix B). The core, load-bearing claims, however, are only partially supported.

  1. "Superior performance over existing state-of-the-art" (abstract, Sec. 5.2). The averaged numbers in Tables 1–2 show ViP2-CLIP is competitive — it is best-on-average on some metric/benchmark pairs and second or worse on others (e.g., MVTec-AD image-level, all three medical image-level datasets, CVC-ColonDB pixel-level where FAPrompt or AdaCLIP lead). Gaps to the runner-up are typically 0.1–1.0%, which is well inside the plausible seed/run variation for these benchmarks. Without seed-level reporting (main tables have no error bars), the ordering between ViP2-CLIP, FAPrompt, AdaCLIP, and AA-CLIP cannot be treated as significant. The abstract's wording should be softened to "competitive/on-par with state-of-the-art."
  2. "UTPA resolves the optimization conflict of dual-branch alignment" (Sec. 4.2, Table 5). The demonstration only compares against a self-built CLIP_DUAL baseline on a frozen backbone with learnable prompts — not against the published dual-branch methods (AnomalyCLIP, APRIL-GAN) that the claim is aimed at. It is therefore not possible to attribute their image/pixel trade-offs to the dual-branch architecture rather than to other design choices. A more controlled ablation — e.g., removing only the global-alignment branch from AnomalyCLIP, or adding it back to ViP2-CLIP — would make the argument causal rather than suggestive.
  3. "Image-conditioned prompts reduce sensitivity to class-name wording" (Sec. 1, Appendix A). The sensitivity experiment is conducted on WinCLIP and AdaCLIP, showing their fragility; however, ViP2-CLIP is never subjected to the analogous stress test (e.g., perturbing the "good"/"damaged" adjectives is done in Table 6, but that is a different axis from class-name wording, and no direct head-to-head under label perturbation is reported). The motivation is therefore argued by elimination rather than by evidence.
  4. "ViP-FGP refines prompts using local visual cues" (Sec. 4.1). The argument rests on Figure 3 (attention-map visualizations) and Table 4 (ablation). Figure 3 is qualitative; Table 4's FGP-only row actually hurts image-level metrics substantially on both MVTec AD (74.1 → 66.1 AUROC) and VisA (60.2 → 60.1), suggesting FGP is beneficial only in combination with other components — a nuance that is not discussed.

  Together, these gaps mean the paper's strongest claims are supported by suggestive rather than conclusive evidence. The method works and is practical, but the paper argues "SOTA + principled fix to a known problem" at a level of certainty the experiments do not reach.

**Requested Changes:**

1. Provide more evidence about the "superior performance over existing state-of-the-art" claim – now it looks as  if the proposed method has slightly competitive/on-par performance with small average-level gains and a favorable efficiency profile. The current wording materially overstates the empirical evidence.
2. Report main-table results with multiple seeds and give standard deviations or confidence intervals. Without these, the 0.1–1.0% gaps that separate ViP2-CLIP from FAPrompt/AdaCLIP/AA-CLIP cannot be interpreted as differences in method quality.
3. Make Table 5 (UTPA vs. dual-branch) causally interpretable. Either (a) re-run AnomalyCLIP / APRIL-GAN after replacing only their alignment strategy with UTPA, keeping everything else fixed, or (b) add UTPA as an ablation on top of ViP2-CLIP's own architecture while varying only the alignment target. The current CLIP_DUAL baseline is too far from the published dual-branch methods to support the "resolves optimization conflict" claim.
4. Add a head-to-head quantitative sensitivity analysis connecting Appendix A with the main contribution: measure PRO/AUROC variance for WinCLIP, AdaCLIP, AnomalyCLIP, FAPrompt, and ViP2-CLIP under the same class-name synonym swaps. This directly tests the paper's motivational claim that image-conditioned prompts are more label-robust.
5. In Table 3, specify the full training configuration used for each method (batch size, image size, number of epochs, precision, whether encoder weights were fine-tuned) and whether the numbers are reproduced from the papers or  measured in a unified setup. As written, training-time differences could reflect protocol differences rather than architectural efficiency.
6. Discuss, with citations, prior uses of Top-K patch pooling for image-level scoring in anomaly detection (the MIL literature, e.g., PatchCore-style aggregations, and the WSOL literature). The current text frames UTPA's aggregator as novel without acknowledging these precedents.

---

> ### Author Response · Authors · 2026-05-21
>
> We sincerely thank you for your insightful comments, which have helped improve the clarity and quality of our manuscript. Below, we provide point-by-point responses and summarize the corresponding revisions made in the updated version. We would be happy to provide any additional information if needed and sincerely look forward to your further feedback.
>
> **Note: In the revised version, all modifications are highlighted in blue.**
>
> > comments 1: Provide more evidence about the "superior performance over existing state-of-the-art" claim. The current wording materially overstates the empirical evidence.
>
> We agree that the original manuscript overstated the empirical evidence in several places. The previous claim of “superior performance over state-of-the-art methods” was primarily based on average improvements, favorable efficiency, and strong pixel-level PRO performance. However, as correctly noted, our method does not consistently outperform existing approaches across all datasets and metrics.
>
> Accordingly, we have revised the manuscript to adopt a more precise and balanced characterization of the results. Specifically, we replaced claims of “superior performance” with “competitive performance” and “particular strengths in pixel-level localization,” which more accurately reflect the empirical findings. These revisions have been applied consistently throughout the Abstract, Section 5.2, and the Conclusion.
>
> In addition, we refined several overly strong expressions (e.g., “resolves” and “addresses”) with more conservative wording to better align with the experimental evidence. Related revisions were incorporated into the Introduction, Related Work, and Section 5.3.
>
> > comments 2: Report main-table results with multiple seeds and give standard deviations or confidence intervals. Without these, the 0.1–1.0% gaps that separate ViP2-CLIP from FAPrompt/AdaCLIP/AA-CLIP cannot be interpreted as differences in method quality.
>
> Following your suggestion, we additionally conducted three-seed experiments for ViP$^{2}$-CLIP and report the mean and standard deviation across all industrial and medical benchmarks. Since several baseline methods only provide official inference checkpoints or single-run reported results, reproducing complete multi-seed evaluations for all competing methods is not always feasible. Therefore, our additional experiments mainly aim to verify that the observed performance trends are reproducible across different random initializations rather than arising from incidental noise.
>
> As shown in Tables 1 and 2, ViP$^{2}$-CLIP consistently exhibits lower variance across different datasets and evaluation metrics, indicating stable optimization behavior and reliable performance. Importantly, the observed standard deviations are substantially smaller than the overall performance trends reported in the main tables, suggesting that the improvements are not caused by random fluctuations.
>
> We have revised the manuscript to present the empirical findings in a more statistically grounded manner. Specifically, the complete results have been added to the revised manuscript as Tables 10–11 in Appendix C.1. We also updated the captions of Tables 1–2 and added corresponding discussions in Sec. 5.2 to explicitly reference these statistical evaluations.
>
> **Table 1. Multi-seed evaluation of ViP²-CLIP on industrial benchmarks**
>
> | Dataset       | Image-level (AUROC, AP, F1)                          | Pixel-level (AUROC, PRO, F1)                         |
> |---------------|------------------------------------------------------|-------------------------------------------------------|
> | MVTec AD      | 91.5 ± 0.18, 96.2 ± 0.21, 92.1 ± 0.16               | 90.7 ± 0.19, 87.0 ± 0.13, 43.2 ± 0.37                |
> | VisA          | 88.5 ± 0.11, 90.3 ± 0.08, 84.9 ± 0.13               | 95.5 ± 0.08, 92.3 ± 0.31, 34.0 ± 0.21                |
> | MPDD          | 79.7 ± 0.21, 84.4 ± 0.27, 82.5 ± 0.19               | 97.1 ± 0.03, 92.5 ± 0.41, 35.8 ± 0.24                |
> | KSDD          | 98.1 ± 0.09, 95.8 ± 0.28, 93.1 ± 0.18               | 98.5 ± 0.18, 96.1 ± 0.17, 53.1 ± 0.19                |
> | BTAD          | 95.1 ± 0.26, 98.3 ± 0.19, 94.6 ± 0.17               | 95.9 ± 0.28, 86.0 ± 0.46, 52.5 ± 0.22                |
> | DAGM          | 98.5 ± 0.01, 94.4 ± 0.27, 92.6 ± 0.11               | 97.3 ± 0.04, 95.4 ± 0.06, 61.4 ± 0.37                |
> | DTD-Synthetic | 95.6 ± 0.22, 98.1 ± 0.11, 94.4 ± 0.28               | 99.2 ± 0.19, 96.3 ± 0.38, 67.6 ± 0.31                |
> | **AVERAGE**   | **92.4 ± 0.16, 93.9 ± 0.23, 90.6 ± 0.19**          | **96.4 ± 0.11, 92.2 ± 0.30, 49.7 ± 0.28**            |

---

> > ### Author Response · Authors · 2026-05-21
> >
> > **Table 2. Multi-seed evaluation of ViP²-CLIP on medical benchmarks**
> >
> > | Dataset       | Image-level (AUROC, AP, F1)                          | Pixel-level (AUROC, PRO, F1)                         |
> > |---------------|------------------------------------------------------|-------------------------------------------------------|
> > | HeadCT        | 94.3 ± 0.11, 94.0 ± 0.02, 88.6 ± 0.28               | —                                                     |
> > | BrainMRI      | 95.5 ± 0.10, 96.5 ± 0.26, 92.4 ± 0.12               | —                                                     |
> > | Brain35H      | 96.1 ± 0.08, 96.2 ± 0.17, 90.4 ± 0.26               | —                                                     |
> > | ISIC          | —                                                    | 90.2 ± 0.02, 82.7 ± 0.22, 74.1 ± 0.31                |
> > | CVC-ColonDB   | —                                                    | 82.9 ± 0.15, 74.9 ± 0.24, 36.6 ± 0.26                |
> > | CVC-ClinicDB  | —                                                    | 86.5 ± 0.13, 71.9 ± 0.18, 45.1 ± 0.17                |
> > | Kvasir        | —                                                    | 82.1 ± 0.19, 48.0 ± 0.21, 47.5 ± 0.20                |
> > | **AVERAGE**   | **95.3 ± 0.10, 95.6 ± 0.21, 90.5 ± 0.26**          | **85.4 ± 0.13, 69.4 ± 0.20, 50.8 ± 0.23**            |
> >
> >
> > > comments 3: Make Table 5 (UTPA vs. dual-branch) causally interpretable.
> >
> > Our initial CLIP_DUAL setting was designed to isolate the effect of the alignment strategy under a simplified framework by freezing the CLIP encoder and optimizing only the learnable prompts. However, as pointed out, this setting does not adequately verify the transferability of UTPA to existing CLIP-based ZSAD methods. Following your suggestion, we additionally conducted controlled experiments by replacing only the original dual-branch alignment strategy in AnomalyCLIP with UTPA, while keeping all other architectures and training settings unchanged.
> >
> > As shown in Table 3, in AnomalyCLIP, replacing conventional dual-branch alignment with UTPA consistently improves both image-level detection and pixel-level localization performance on MVTec AD and VisA. In addition, the ablation results in Table 4 (Rows 4 and 7) further demonstrate the effectiveness of UTPA within our ViP$^{2}$-CLIP architecture itself. Together, these results provide a more controlled and causally interpretable validation of the transferability and effectiveness of the proposed alignment strategy.
> >
> > Based on these additional experiments, we have revised the discussion in Sec. 5.3 (“UTPA for Mitigating Optimization Conflicts”) and updated Table 5 to provide a more rigorous and balanced analysis.
> >
> > **Table 3. Performance comparison between dual-branch alignment and the UTPA strategy**
> >
> > | Module               | MVTec AD (Pixel-level)       | MVTec AD (Image-level)       | VisA (Pixel-level)           | VisA (Image-level)           |
> > |----------------------|------------------------------|------------------------------|------------------------------|------------------------------|
> > | CLIP_DUAL            | (69.6, 24.6, 12.0)           | (88.6, 94.9, 91.3)           | (93.3, 82.4, 24.2)           | (79.3, 82.4, 79.0)           |
> > | CLIP_UTPA            | (89.6, 83.7, 37.4)           | (85.0, 93.3, 88.8)           | (94.4, 88.4, 26.2)           | (83.8, 86.7, 81.4)           |
> > | AnomalyCLIP          | (91.1, 81.4, 39.1)           | (91.3, 96.4, 92.7)           | (95.5, 86.7, 28.3)           | (82.0, 85.3, 80.4)           |
> > | AnomalyCLIP_UTPA     | (91.3, 83.2, 41.3)           | (91.5, 96.7, 93.1)           | (95.5, 88.6, 30.1)           | (84.4, 87.5, 82.9)           |
> >
> > > comments 4: Add a head-to-head quantitative sensitivity analysis connecting Appendix A with the main contribution: measure PRO/AUROC variance for ZSAD methods.
> >
> > Thank you for this valuable suggestion. We believe there may be a slight misunderstanding regarding our prompt formulation. Unlike prior CLIP-based ZSAD methods that explicitly rely on category names[1][2][3], ViP$^{2}$-CLIP replaces the class-name position with visually guided learnable tokens. Consequently, the framework does not require manually specified category labels during inference, and the corresponding PRO/AUROC variance under category-name synonym swaps is theoretically zero by design.
> >
> > Under this formulation, the more meaningful robustness analysis concerns the remaining manually specified textual priors, namely the descriptive adjectives (e.g., “good” and “damaged”). Following this motivation, we already conducted the adjective replacement analysis in Table 6. The results show highly consistent performance across different adjective choices, demonstrating reduced sensitivity to handcrafted prompt wording and improved robustness to textual variations.
> >
> > To avoid ambiguity, we have further clarified the distinction between category-name dependence and adjective-level textual priors in the revised Appendix A.

---

> > > ### Author Response · Authors · 2026-05-21
> > >
> > > > comments 5: Table 3 lacks consistent training configuration across methods; reported efficiency may be confounded by differing experimental setups.
> > >
> > > Thank you for this valuable suggestion. All methods were evaluated under the same hardware environment, inference protocol, and input resolution to minimize external discrepancies. Since existing CLIP-based ZSAD methods adopt different optimization strategies and training recipes, we followed the official implementations and default settings of all compared methods, with training configurations summarized in Table 4 below.
> > >
> > > Following your suggestion, we have revised the "Computational Efficiency Analysis" in Sec. 5.2 and added a detailed training-configuration comparison in the revised Table 8 (Appendix B.2) for a more rigorous and transparent efficiency analysis.
> > >
> > >
> > > **Table 4. Training configurations used in the computational efficiency comparison**
> > >
> > > | Method           | Image Size | Batch Size | Epochs | Precision | Encoder Fine-tuning |
> > > |------------------|------------|------------|--------|-----------|---------------------|
> > > | AnomalyCLIP      | 518        | 8          | 15     | FP32      | ✓                   |
> > > | AdaCLIP          | 518        | 1          | 5      | FP32      | ✓                   |
> > > | AA-CLIP          | 518        | 2          | 25     | FP32      | ✓                   |
> > > | FAPrompt         | 518        | 8          | 15     | FP32      | ✓                   |
> > > | ViP²-CLIP        | 518        | 8          | 10     | FP32      | —                   |
> > >
> > >
> > > > comments 6: Discuss, with citations, prior uses of Top-K patch pooling for image-level scoring in anomaly detection.
> > >
> > > Following your suggestion, we have revised Sec. 4.2 (third paragraph) to better position our aggregation strategy within prior regional pooling paradigms, including multiple instance learning (MIL)[4], weakly supervised object localization (WSOL), and patch-based anomaly detection (e.g., PatchCore[6]).
> > >
> > > These approaches share a common principle: image-level predictions are driven by a subset of discriminative local instances rather than global averages. MIL identifies key instances within a bag, WSOL localizes regions via spatial response aggregation (e.g., CAM[5]) under image-level supervision, and PatchCore derives anomaly scores from the most abnormal patch features. Motivated by this shared design, our method aggregates image-level anomaly scores from the most anomalous patch responses, preserving localized evidence while improving global discrimination.
> > >
> > > > comments 7: ViP-FGP contribution not fully supported by ablation behavior.
> > >
> > > Thank you for this insightful comment. We agree that Table 4 shows a drop in image-level AUROC when ViP-FGP is used alone (MVTec AD: 74.1 → 66.1; VisA: 60.2 → 60.1), which was not sufficiently discussed in the original manuscript.
> > >
> > > This occurs because ViP-FGP is designed to inject strong local visual sensitivity via cross-modal interaction. When used in isolation, this may over-emphasize local irregularities and introduce noise into global semantic aggregation, leading to reduced image-level consistency. Importantly, ViP-FGP is not intended as a standalone scoring module, but as a complementary component to ViP-VCA, which provides stable object-level semantic priors.
> > >
> > > Their combination enables a balance between global semantic stability and local defect sensitivity, leading to improved overall performance. We have clarified this complementary relationship in Sec. 5.3.
> > >
> > >
> > > **References**
> > >
> > > [[1]](https://openaccess.thecvf.com/content/CVPR2023/html/Jeong_WinCLIP_Zero-Few-Shot_Anomaly_Classification_and_Segmentation_CVPR_2023_paper.html) Jeong, J., et al. WinCLIP: Zero-/Few-Shot Anomaly Classification and Segmentation. CVPR, 2023.
> > >
> > > [[2]](https://link.springer.com/chapter/10.1007/978-3-031-73039-9_8) Cao, Y., et al. AdaCLIP: Adapting CLIP with Hybrid Learnable Prompts for Zero-Shot Anomaly Detection. ECCV, 2024.
> > >
> > > [[3]](https://arxiv.org/abs/2501.03786) Li, C., et al. KAnoCLIP: Zero-Shot Anomaly Detection through Knowledge-Driven Prompt Learning. arXiv, 2025.
> > >
> > > [[4]](https://proceedings.mlr.press/v80/ilse18a.html) Ilse, M., et al. Attention-based Deep Multiple Instance Learning. ICML, 2018.
> > >
> > > [[5]](https://openaccess.thecvf.com/content_cvpr_2016/html/Zhou_Learning_Deep_Features_CVPR_2016_paper.html) Zhou, B., et al. Learning Deep Features for Discriminative Localization. CVPR, 2016.
> > >
> > > [[6]](https://openaccess.thecvf.com/content/CVPR2022/html/Roth_Towards_Total_Recall_in_Industrial_Anomaly_Detection_CVPR_2022_paper.html) Roth, K., et al. Towards Total Recall in Industrial Anomaly Detection. CVPR, 2022.

---

> > > > ### Comment · Reviewer_A6pY · 2026-05-21
> > > > **Response to authors**
> > > >
> > > > I appreciate the authors for the responses and additional experiments. Here's my thoughts about them (before making a final decision):
> > > >
> > > > 1. Tables 1–2 confirm that ViP2-CLIP itself is stable across seeds (std=0.01~0.46), which is reassuring on the "is the result reproducible" aspect. However, I'm still concerned about whether the 0.1–1.0% gaps between ViP2-CLIP from FAPrompt / AdaCLIP / AA-CLIP are statistically meaningful. The new tables actually make these ties more visible, not less. For example, on the medical image-level average ViP2-CLIP is 95.3 ± 0.10 versus FAPrompt 95.7.  I'd suggest the manuscript explicitly state where these ties exist (and maybe why, because of data distribution perhaps?), that the methods are statistically indistinguishable on those particular benchmarks, rather than simply implying ViP2-CLIP wins.
> > > >
> > > > 2. Table 3 (UTPA grafted onto AnomalyCLIP). This is the strongest new evidence and the one that genuinely improves the paper. The MVTec-AD gains are small (image AUROC +0.2, F1 +0.4; pixel PRO +1.8, F1 +2.2) and would benefit from seeds, but the VisA gains (image AUROC +2.4, AP +2.2, F1 +2.5; pixel PRO +1.9) are large enough to look real. I appreciate that.
> > > >
> > > > 3. Table 4 (training-config table). Useful for transparency, but it does not show better performance, and it does change how I read Table 3 in the original paper. Making explicit that all four compared methods fine-tune the CLIP encoder while ViP2-CLIP does not is honest, but it means the training-time and parameter-count gap reflects a different fine-tuning regime as much as a different architecture. The "lightweight" framing is still accurate, but I'd encourage the manuscript to phrase the efficiency comparison as "ViP2-CLIP achieves competitive accuracy without encoder fine-tuning, which is the dominant source of its lower training cost" rather than presenting the gap as purely architectural.
> > > >
> > > > 4. Regarding "head-to-head quantitative sensitivity analysis", I'm not asking to perturb something inside ViP2-CLIP's own prompt. It's true that ViP2-CLIP has no class-name slot, so obviously its variance under category-name swaps is 0. I'm asking for a comparative figure, on the same images and the same metrics, that quantifies the gap between: (a) the worst-case behavior of class-name-dependent baselines under realistic label ambiguity, and (b) ViP2-CLIP's single fixed operating point. ViP2-CLIP's role in this experiment is the reference line, and the variable is the baseline. Appendix A shows that WinCLIP can swing by ~10% and AdaCLIP by up to 47% PRO when "carpet" is replaced with "floor mat." The abstract and introduction then argue that ViP2-CLIP is the right answer in "scenarios with ambiguous or privacy-constrained category labels." The paper currently shows the problem (Appendix A) and proposes the image-conditioned prompts as solution, but never measures the actual delta between them. It would help the paper's logic, if the motivation and the contribution are connected by the comparison I asked for.
> > > >
> > > >
> > > > 5. I'd still appreciate a response on the Broader Impact Statement, which the authors' reply somehow misses out.

---

> ### Author Response · Authors · 2026-05-22
>
> Thank you for your constructive feedback. Your insightful suggestions have helped make our manuscript more rigorous and complete. Below, we provide a point-by-point response to your concerns and summarize the corresponding revisions made in the updated version. We would be happy to provide any additional information if needed and sincerely look forward to your further evaluation.
>
> > comments 1:  Regarding the statistical significance of the reported gains.
>
> Thank you for this valuable suggestion. In the revised manuscript, we toned down overly strong claims and explicitly clarified where the compared methods exhibit statistically comparable performance. Specifically, we added discussions in Section 5.2 noting that ViP$^2$-CLIP shows relatively clearer gains on industrial benchmarks such as VisA, MPDD, and BTAD, while the remaining datasets exhibit narrower performance margins (comparable results to recent competitive approaches). We further emphasize that the proposed framework mainly provides competitive and stable detection performance together with consistently strong pixel-level localization capability across diverse industrial benchmarks.
>
> For medical benchmarks, we additionally clarified that ViP$^2$-CLIP achieves more noticeable improvements on datasets such as BrainMRI and CVC-ClinicDB, while the relatively limited gains on several remaining datasets are likely caused by the substantial domain gap between industrial auxiliary data and medical target domains. Such distribution discrepancies may weaken the generalizability of the transferred visual-conditioned prompts for cross-domain semantic alignment. We further included supplementary analyses in Appendix C.4, showing that incorporating medically related auxiliary data during finetuning can further improve prompt alignment quality and localization performance.
>
> This observation further suggests that by injecting global visual feature into the text prompts, our learnable visual-guided prompting mechanism enhances cross-modal alignment. However, when the distribution gap between global visual features becomes substantial, finetuning with domain-specific auxiliary data appears to be a more effective strategy. Thank you again for these insightful suggestions.
>
> > comments 2: Regarding Table 3 (UTPA grafted onto AnomalyCLIP).
>
> Combined with the results of other baselines in the main tables, we believe that the image-level detection performance on MVTec AD may already be approaching saturation for recent CLIP-based ZSAD methods, making further improvements comparatively challenging. As a result, the gains introduced by UTPA on image-level metrics are relatively marginal, and in some cases nearly unchanged.
>
> To make the discussion more precise and balanced, we revised the description in Section 5.3 (“UTPA for Mitigating Optimization Conflicts”). Specifically, we replaced: “The resulting variant, AnomalyCLIP_UTPA, consistently improves both localization and detection performance across MVTec AD and VisA datasets.” with: “The resulting variant, AnomalyCLIP_UTPA, achieves notable improvements in pixel-level localization on MVTec AD and in both image-level detection and pixel-level localization on VisA.”
>
> > comments 3: Regarding Table 4 (training-config table).
>
> Following your suggestion, we added an additional column in Table 3 indicating whether the CLIP encoder is fine-tuned for each compared method. We also refined the discussion in Section 5.2 (“Computational Efficiency Analysis”) to clarify that the efficiency advantage of ViP$^2$-CLIP mainly arises from achieving competitive performance without CLIP encoder fine-tuning, which is the dominant factor contributing to its lower training cost and parameter overhead.

---

> ### Author Response · Authors · 2026-05-22
>
> > comments 4: Regarding "head-to-head quantitative sensitivity analysis".
>
> Thank you for the detailed clarification. We fully agree that the key issue is not the internal prompt stability of ViP$^2$-CLIP itself, but rather the quantitative gap between the worst-case behavior of class-name-dependent baselines under realistic label ambiguity and the fixed operating point of our label-agnostic framework. To better close this motivation-to-contribution loop, we added a new comparative analysis in Appendix A.2.
>
> Specifically, following the ambiguity analysis in Appendix A.1, we further evaluate both the training-free WinCLIP and the finetuned AdaCLIP under semantically similar category-name substitutions on three representative MVTec AD categories: bottle, zipper, and carpet. (AnomalyCLIP and FAPrompt are not included in this evaluation, as they are inherently independent of category-name information.) The evaluated substitutions include (bottle, cap, jar), (zipper, zip, fastener), and (carpet, rug, floor_mat), where the substituted labels remain highly semantically related in order to simulate realistic industrial scenarios with ambiguous or inconsistent category descriptions.
>
> As shown in the newly added Fig. 13, both WinCLIP and AdaCLIP exhibit substantial performance fluctuations across different category-name substitutions, while ViP$^2$-CLIP serves as a stable and more competitive reference baseline under the same evaluation setting. In particular, for the \textit{zipper} category, AdaCLIP achieves relatively strong performance when using the synonym zip'', with pixel-level PRO and F1 approaching our baseline. However, when using zipper'' or ``fastener'', the pixel-level F1 drops dramatically (e.g., only 2.43), leading to highly blurred anomaly boundaries and unreliable localization results.
>
> These cases indicate that, even when using semantically very similar category descriptions, class-name-dependent methods may still struggle to consistently achieve their optimal performance and can exhibit substantial instability under ambiguous labeling conditions. In contrast, the proposed image-conditioned prompting mechanism provides a comparatively stable and competitive localization baseline without relying on manually specified class names. We believe this property is particularly valuable for practical industrial inspection scenarios where category descriptions may be uncertain, inconsistent, or unavailable.
>
> > comments 5: Regarding the Broader Impact Statement.
>
> Sorry for the earlier omission. We have added a new “Ethical and Deployment Considerations” section at the end of the manuscript. In this section, we explicitly discuss the use of publicly available anonymized datasets and the potential risks of real-world medical deployment, including possible prediction and localization errors. We further clarify that ViP$^2$-CLIP is intended for research purposes only and should be deployed with appropriate human expert verification and clinical oversight.

---

> > ### Comment · Reviewer_A6pY · 2026-05-25
> > **Response to authors**
> >
> > I appreciate the authors for engaging with the discussion process seriously. The revised version presents the contribution with better support, and I see the majority of my concerns addressed.

---

### Author Response · Authors · 2026-05-25

Dear AC and Reviewers,

We sincerely thank all reviewers for their valuable and constructive feedback, which has helped improve the clarity, rigor, and completeness of the paper. In the following, we provide a structured summary of all comments together with our corresponding responses and revisions, to facilitate a clearer understanding of the changes made in the revised manuscript.

> Comment 1: Regarding the overstatement of “superior performance over existing state-of-the-art methods”.

To address this concern, we have revised the manuscript to ensure a more accurate and balanced presentation of the results. Specifically:

- We replaced overly strong claims such as “superior performance” with “competitive performance” and “particular strengths in pixel-level localization”;
- We systematically reviewed the manuscript to remove or soften overstated expressions (e.g., “resolves”, “addresses”) where necessary;

These revisions have been applied throughout the manuscript, including the Abstract, Introduction, Related Work, Section 5.2, Section 5.3, and Conclusion.

> Comment 2: Regarding the absence of multi-seed results in the main table.

We have conducted additional experiments with three different random seeds for our method, and report the corresponding results in Appendix C.1 (Tables 10–11). The results show that ViP$^2$-CLIP exhibits low variance (≤ 0.3% across key metrics), indicating that the proposed method is stable and reproducible under different initialization conditions.

In addition, based on these, we have revised Section 5.2 to provide a more fine-grained interpretation of the main results across the 14 benchmarks. Specifically, we now explicitly distinguish performance trends across dataset groups:

- ViP$^2$-CLIP shows clear improvements on industrial datasets (VisA, MPDD, BTAD);
- It maintains competitive performance on the remaining industrial subsets;
- It also achieves notable gains on medical datasets (BrainMRI and CVC-ClinicDB).

We further provide a more cautious discussion of cases where improvements are marginal, attributing them to differences in visual characteristics and domain shifts across datasets, where the visual-conditioned adapter may have limited additional generalization gains.

These revisions have been applied in Section 5.2 and Appendix C.1.

> Comment 3: Regarding the concern on the causal interpretability of Table 5 (UTPA vs. dual-branch alignment).

We conducted additional controlled experiments by replacing only the original dual-branch alignment strategy in AnomalyCLIP with our proposed UTPA module. The results provide a more controlled and causally interpretable validation of the transferability and effectiveness of the proposed alignment strategy.

These revisions have been applied in Section 5.3 (“UTPA for Mitigating Optimization Conflicts”).

> Comment 4: Regarding the lack of head-to-head sensitivity analysis under class-name synonym substitutions.

To better close this motivation-to-contribution loop, we extend Appendix A.2 with a head-to-head sensitivity analysis following the ambiguity study in Appendix A.1. Specifically, we evaluate WinCLIP and AdaCLIP under semantically similar category-name substitutions on three representative MVTec AD categories (bottle, zipper, carpet), and further examine the performance gain of ViP$^2$-CLIP under ambiguous label conditions. The results show that ViP$^2$-CLIP maintains more stable and competitive performance under the same evaluation setting, while class-name-dependent methods exhibit noticeable performance fluctuations across semantically equivalent substitutions.

These revisions have been applied in Appendix A.2 and the Introduction Section.

> Comment 5: Regarding the interpretability of the efficiency comparison in Table 4.

To further improve reproducibility and transparency, we provide a detailed breakdown of training configurations for all compared methods in Appendix B.2. We additionally revise Table 3 to explicitly indicate whether the CLIP encoder is fine-tuned for each compared method, and clarify that the efficiency gap mainly stems from differences in training paradigms, as ViP$^2$-CLIP does not fine-tune the CLIP encoder.

These revisions have been applied in Section 5.2 (Computational Efficiency Analysis) and Appendix B.2.

> Comment 6: Regarding prior work on Top-K patch pooling for image-level scoring.

In response, we revise Section 4.2 to incorporate a discussion of related regional pooling paradigms, including multiple instance learning (MIL), weakly supervised object localization (WSOL), and patch-based anomaly detection methods such as PatchCore. We clarify that our Top-K patch pooling follows this general design principle by aggregating anomaly scores from the most informative patch responses, thereby preserving localized evidence while improving global-level discrimination.

These revisions have been applied in Section 4.2.

---

> ### Author Response · Authors · 2026-05-25
>
> > Comment 7: Regarding the role of ViP-FGP and its ablation behavior.
>
> In response, we clarify in Section 5.3 that the performance drop observed when ViP-FGP is used in isolation stems from its design. ViP-FGP introduces strong local visual sensitivity through cross-modal interaction, which may over-emphasize fine-grained cues and reduce global image-level consistency. We further emphasize that ViP-FGP is designed to work jointly with ViP-VCA. Their combination improves overall performance by balancing global semantic stability and local defect sensitivity.
>
> These revisions have been applied in Section 5.3 (Module Ablation).
>
> > Comment 8: Regarding expanding synonym-swap analysis to full MVTec AD and reporting aggregate statistics.
>
> In response, we extend the synonym-substitution experiments from a subset of categories to the full MVTec AD dataset. The results are reported in Appendix A.1 (Figures 8–12 and Table 7), together with aggregate statistics including the mean and standard deviation of performance changes across all categories and synonym variants. The results consistently show that image-level metrics remain relatively stable under semantic label substitutions, whereas pixel-level localization exhibits significantly larger variations across categories. This further highlights the sensitivity of class-name-dependent methods to label ambiguity.
>
> These revisions have been applied in Appendix A.1.
>
> > Comment 9: Regarding comparison with concurrent and related CLIP-based ZSAD methods.
>
> In response, we clarify that GenCLIP is excluded due to the lack of publicly available code, and ACD-CLIP has not yet been officially released. VCP-CLIP is primarily designed for segmentation-only settings, making it less directly comparable to our unified detection and localization framework. We further organize existing CLIP-based ZSAD methods into four representative paradigms: (1) training-free prompt ensembles, (2) class-name-dependent encoder fine-tuning methods, (3) object-agnostic prompt learning approaches, and (4) vision-guided learnable prompting methods. Our experimental evaluation already includes representative methods from each category, ensuring a comprehensive and fair comparison.
>
> These revisions clarify the positioning of ViP$^2$-CLIP relative to existing paradigms and have been reflected in the revised Related Work section.
>
> > Comment 10: Regarding deeper limitation analysis and performance gaps across datasets/metrics.
>
> In response, we add a detailed analysis in Appendix C.5, focusing on cases where ViP$^2$-CLIP underperforms strong baselines on image-level metrics for certain datasets (e.g., MVTec-AD and VisA). We identify the fixed Top-$K$ aggregation strategy in UTPA as the primary factor. While Top-$K$ pooling effectively aligns detection and localization by emphasizing the most suspicious patches, it may introduce noise in cases with small or low-contrast anomalies, where non-defective regions can be included in the aggregation. This can weaken the global anomaly signal and reduce image-level sensitivity. These results highlight a trade-off in the proposed aggregation strategy and suggest that adaptive aggregation across spatial scales is a promising direction for future improvement.
>
> The corresponding analysis has been added in Appendix C.5.
>
> > Comment 11: Regarding sensitivity of prompt token number and UTPA layer selection.
>
> We clarify that these implementation details are already included in the revised manuscript. For the prompt token configuration, the analysis in Appendix C.2 shows that increasing prompt length does not consistently improve performance, and overly long prompts may even hurt generalization. For UTPA, the layer-wise analysis in Section 5.3 (Visualization at Different Layers) shows that different encoder layers provide complementary information, and aggregating multiple layers leads to more robust anomaly representations.
>
> These clarifications have been made in Appendix C.2 and Section 5.3 (Visualization at Different Layers) of the revised manuscript.
>
> > Comment 12: Regarding potential future directions.
>
> We have noted that incorporating spatial priors and adaptive aggregation strategies into the unified text-patch alignment framework is a promising direction for improving robustness in such challenging scenarios.
>
> These revisions have been applied in Appendix C.5.

---

> > ### Author Response · Authors · 2026-05-25
> >
> > > Comment 13: Regarding the performance gap between industrial and medical datasets.
> >
> > In response, we add a detailed analysis in Appendix C.4, including feature representation analysis and attention visualizations (Figure 15), to investigate the performance gap between industrial and medical datasets. The results show that the gap is primarily driven by domain differences in anomaly characteristics. Industrial anomalies are typically structural or textural, whereas medical anomalies often exhibit low contrast and ambiguous boundaries, leading to distinct feature distributions and reduced transferability of models trained on industrial data. Visualization results further indicate that incorporating domain-relevant medical data improves attention localization on lesion regions, while training only on industrial data leads to less focused and less discriminative attention in medical scenarios. Overall, these findings highlight the importance of domain-relevant auxiliary data for improving cross-domain generalization in prompt-based ZSAD.
> >
> > The corresponding analysis has been added in Appendix C.4.
> >
> >
> > We sincerely thank all reviewers for their valuable and constructive feedback.

---

### Decision · Action_Editor_kGSJ · 2026-07-04

**Recommendation:** Accept as is

**Additional Comments:**

This paper proposes ViP²-CLIP method, which is a clean, lightweight, label-agnostic CLIP adaptation for ZSAD, with one well-supported architectural insight (UTPA -- replacing dual-branch alignment with a unified text–patch alignment plus Top-K pooling) and a quantified demonstration that the label-agnostic prompting design is more stable than class-name baselines under realistic label ambiguity.

However, the contribution depends on a clean class-name-free baseline and an effective alignment simplification, with the clearest gains in pixel-level localization and in efficiency that stems mainly from not fine-tuning the encoder. Since the individual components are each close to existing ideas, I'd still see the overall contribution as useful and practical.

**Audience:**

Yes

**Audience Explanation:**

Zero-shot anomaly detection is a practically important problem with applications in manufacturing quality control and medical imaging, both areas where labeled anomaly data is scarce or expensive. The paper addresses a real and well-motivated limitation of existing CLIP-based ZSAD methods sensitivity to class-name wording, which is relevant to practitioners deploying these systems. The audience interested in vision-language models, prompt learning, and industrial/medical anomaly detection would find value in the proposed solutions. The breadth of evaluation (14 benchmarks across two domains) also increases the paper's relevance.

**Claims And Evidence:**

Yes

**Claims Explanation:**

The core contribution is now framed: ViP²-CLIP is a clean, lightweight, label-agnostic CLIP adaptation for ZSAD, with one well-supported architectural insight (UTPA -- replacing dual-branch alignment with a unified text–patch alignment plus Top-K pooling) and a quantified demonstration that the label-agnostic prompting design is more stable than class-name baselines under realistic label ambiguity. The wording overreach in the original draft has been recalibrated to match what the experiments actually show.